# Combined Earth observations reveal the sequence of conditions leading to a large algal bloom in Lake Geneva
Abolfazl Irani Rahaghi [1,2] ✉, Daniel Odermatt [1,2], Orlane Anneville [3], Oscar Sepúlveda Steiner [4,5], Rafael Sebastian Reiss [6], Marina Amadori [7], Marco Toffolon [8], Stéphan Jacquet [3], Tristan Harmel [9], Mortimer Werther[1], Frédéric Soulignac [10], Etienne Dambrine[3], Didier Jézéquel [3,11], Christine Hatté [12,13], Viet Tran-Khac[3], Serena Rasconi [3], Frédéric Rimet[3] & Damien Bouffard [4,14]

Freshwater algae exhibit complex dynamics, particularly in meso-oligotrophic lakes with sudden and dramatic increases in algal biomass following long periods of low background concentration. While the fundamental prerequisites for algal blooms, namely light and nutrient availability, are well-known, their specific causation involves an intricate chain of conditions. Here we examine a recent massive Uroglena bloom in Lake Geneva (Switzerland/France). We show that a certain sequence of meteorological conditions triggered this specific algal bloom event: heavy rainfall promoting excessive organic matter and nutrients loading, followed by wind-induced coastal upwelling, and a prolonged period of warm, calm weather. The combination of satellite remote sensing, in-situ measurements, ad-hoc biogeochemical analyses, and three-dimensional modeling proved invaluable in unraveling the complex dynamics of algal blooms highlighting the substantial role of littoral-pelagic connectivities in large low-nutrient lakes. These findings underscore the advantages of state-of-the-art multidisciplinary approaches for an improved understanding of dynamic systems as a whole.

Phytoplankton blooms in low-nutrient (meso-oligotrophic) water bodies are short-lived and difficult to predict. While promoting and inhibiting conditions for long-term changes in phytoplankton phenology and bloom frequency are known[1,2], the specific triggers of phytoplankton blooms remain enigmatic for most individual events[3,4]. Climate change has been identified as one of the main drivers[1,5,6], yet without compelling correlation between air temperature or precipitation and algal blooms. Such concealed responses highlight the complexity of the system's dynamics and underscores the need to consider a variety of factors when trying to predict short-lived algal blooms.

In September 2021, a massive Uroglena sp. bloom was observed in Lake Geneva (Switzerland/France), a large and deep low-nutrient freshwater lake. Such a massive bloom has been, to the best of our knowledge, unprecedented since end-1900s when phosphorus concentration drastically decreased following the recommendations of the Commission Internationale pour la Protection des Eaux du Léman (CIPEL). This rare, eye-catching event attracted public attention and was vastly covered in regional newspapers and social media, with the urge to explain in near real-time the blooms' source and fate, challenging the traditionally longer study time scale of scientists.

A comprehensive causal understanding of algal blooms necessitates a multidisciplinary approach[7,8] that combines bio-geophysical, ecological, and atmospheric data[3,9,10]. Recent advances in automated workflows for remote sensing data and hydrodynamic models allow reconstruction of driving factors behind short-term algal blooms[11-13] and the underlying mechanisms, including their build-up and dispersion[14]. Here, we integrate an array of Earth observation approaches and tools (in-situ measurements, satellite remote sensing, 3D hydrodynamic and Lagrangian particle tracking) to investigate the formation and spatial dynamics of the exceptional 2021 algal bloom in Lake Geneva. Our multidisciplinary approach provided insightful perspectives into the causation of this exceptional bloom in Lake Geneva, highlighting the role of the "right" timing of the atmospheric forcing and the subsequent hydrodynamic processes. This was supported with statistical analysis of the frequency of the appropriate temporal combination of these forcing conditions in Lake Geneva during 1998–2022.

**Article**

## Results and discussion

### Algal bloom characterization: Insights from satellite remote sensing and in-situ measurements

On September 6, 2021, a Sentinel-2 Multi-Spectral Instrument (MSI) satellite image indicated a widespread bloom in Lake Geneva (Fig. 1), in the form of a prominent three-band reflectance index[15]. From the latter we estimated large regions (>5 km wide) of high phytoplankton chlorophyll-a concentrations ($C_{chl-a}$ > 50 mg m$^{-3}$), and low transparency (Secchi depth $z_{SD}$ < 3 m) with extreme values in the southern shores (Supplementary Fig. 6). Automated high-frequency in-situ profiles of $C_{chl-a}$ and 700 nm backscattering coefficient ($bb_{700nm}$) showed a shift of subsurface chl-a and backscatter maxima towards the surface (~0.5–3 m depth; Supplementary Fig. 7), indicating the development of a surface bloom. The observed extreme values of $z_{SD}$ and $C_{chl-a}$, compared with historical long-term in-situ measurements in Lake Geneva (Supplementary Fig. 8), confirmed the extraordinary intensity of this bloom.

Taxonomic identification revealed extremely high concentrations of Uroglena cells, a motile, colonial mixotrophic phytoplankton species that can feed on bacteria and dissolved organic matter, and maintain dense blooms under nutrient-limited conditions[3,16]. This phytoplankton bloom was monospecific (Supplementary Fig. 9) in all the sampled locations on September 10 (orange circles in Fig. 1), except for SP4 with a low phytoplankton abundance and a more diverse community. In addition, the typical fishy odor and brownish color characteristics of Uroglena blooms[3] were reported during the beginning of September 2021. Uroglena remained dominant until September 13 with a strong spatial heterogeneity of $C_{chl-a}$ (4.4 mg m$^{-3}$ to 22 mg m$^{-3}$) and $z_{SD}$ (1.6–3.2 m) within <1 km proximity of SP1 (Fig. 1).

Sentinel-3 Ocean and Land Color Instrument (OLCI) satellite images (Supplementary Fig. 10) suggest the bloom's onset between September 2 and 4, initially growing along the southern shores and culminating on September 6 (Fig. 1). The observed event can be categorized as a sporadic bloom specified with a rare biomass spike, a high rate of change, and spatially advected patterns[8]. The latter is characterized by striking 1–10-km wide structures throughout the basin, including mushroom-like features in the west and center, and a gyre with filamentary patterns along its edges in the east (Fig. 1). Such patterns (mimicking Van Gogh's famous nuit étoilée painting) indicate that complex hydrodynamics shaped the bloom by

basin-scale advection of biomass. Modeling vertical and horizontal advective processes can help to understand the dynamics of sporadic blooms[8,17,18].

### Hydrodynamic modeling and Lagrangian particle tracking of the bloom

The circulation was simulated using MITgcm[19], with model configurations from Safin et al.[20] (See Supplementary Methods for details of model validation and evaluation). Backward and forward Lagrangian particle tracking[21,22] were then employed to identify transport pathways and determine the origin and fate of the bloom large-scale patterns. Numerical modeling resembled the main observed satellite patterns. Forward trajectories of particles released hourly near the southern shore from September 3 (estimated bloom onset from both satellite and in-situ observations) onwards suggest that the western and central mushroom-like structures originated from these littoral regions (Fig. 2a and Supplementary Movie 1; see also backward tracking results in Supplementary Fig. 11A, B and Supplementary Movies 2−3). In contrast, backward tracking of particles released on September 6 in the eastern bloom structure revealed that pelagic waters had mostly been trapped in this area (Supplementary Fig. 11C and Supplementary Movie 4), with only weak connections to the southern shore and the lake's eastern end.

Empirical Orthogonal Function (EOF) analysis of the modeled flow field indicated an array of surface rotational currents that dominated the lake's circulation and sustained spatial heterogeneities for about four days after the bloom onset (Fig. 2b). Three basin-scale rotational patterns are distinguishable in the first EOF mode going from west to east: (i) relatively weak anticyclonic (clockwise), (ii) strong anticyclonic, and (iii) strong cyclonic (counter-clockwise). Furthermore, a smaller anticyclonic meso-scale eddy appeared in the easternmost part of the lake. The transport induced by these basin-scale circulations connected the littoral to the pelagic waters and maintained lateral heterogeneities. Recent studies have demonstrated that such patterns are characteristic of Lake Geneva's surface circulation, with the details depending on wind forcing and stratification[23,24].

Model results during four days preceding the bloom show that wind-induced coastal upwelling along the main basin's southern shore (Fig. 2c) lifted the southern thermocline by ~10 m, placing it into the photic zone (Fig. 2d). Weaker upwelling was also observed along the northern shore,

**Fig. 1 | The map of the three-band reflectance index in Lake Geneva on September 6, 2021.** The map is calculated from Sentinel-2 MSI remote sensing reflectance products at three red and Near-InfraRed (NIR) wavelengths (665, 704, and 740 nm)[15,90]. This optical index is a proxy for chlorophyll-a concentration ($C_{chl-a}$). The obtained values on the map suggest large regions with approximately $C_{chl-a}$ > 50 mg m$^{-3}$ (Supplementary Fig. 6). The red square, red triangle, red star, and blue diamond indicate the location of the offshore LéXPLORE research platform, the historical SHL2 monitoring site, the near-shore Buchillon station, and a near-lake meteorological station at Changins, respectively. The orange circles show the water sampling locations a few days after the bloom's peak. Rhône (in) and Rhône (out) are the lake's main river inflow and outflow, respectively, with La Dranse (in) being the second largest inflow. The gray dots indicate major cities around the lake. Topographic features on the map are from the Swiss Federal Office of Topography (SwissTopo).

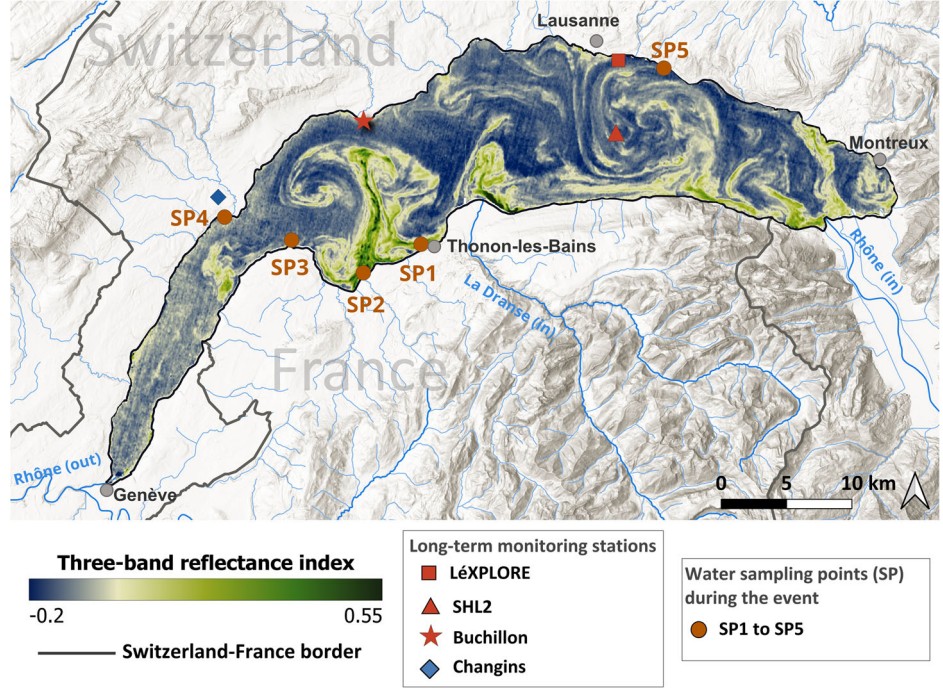

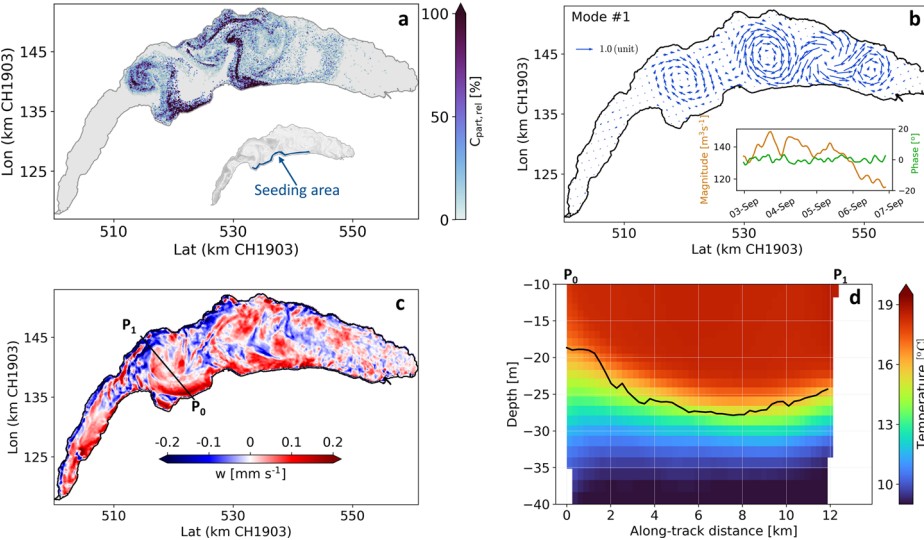

**Fig. 2 | Results from 3D hydrodynamic modeling and Lagrangian particle tracking. a** location of forward-tracked particles on September 6 at 14:30 after hourly releases since September 3, **b** first EOF mode of velocity flux from 0 to 20 m depth, **c** vertical velocity averaged between August 31 to September 3 from 0 to 20 m depth, and **d** temperature profile along $P_0$-$P_1$ transect in **c** during the same period. The blue area in the inset plot in **a** indicates the seeding area where particles were released randomly between 0 and 10 m depth. The inset in **b** shows the temporal variation of magnitude and phase of the first EOF mode. The key for the spatial quiver plot in panel **b** is given on the top left. The first and second (Supplementary Fig. 12) EOF modes determined 62.2% and 12.3% of the total variance of the horizontal flow field, respectively. The solid black line in **d** indicates the thermocline location along the transect.

## Identification of Carbon sources at the origin of the bloom using $^{14}$C

The source of carbon used by the algae to replicate and create the bloom was examined by analyzing radiocarbon content of the bloom ($^{14}$C; Supplementary Table 2). The bloom sampled in shallow water at Séchex littoral (SP2 in Fig. 1) on September 10 was strongly enriched in $^{14}$C ($F^{14}C = 1.225$) compared to atmosphere ($F^{14}C$ just under 1.000 value[25]), an indication of incorporating carbon from terrestrial sources. Total organic matter of soil and subsoil samples from littoral eroded areas were slightly enriched ($F^{14}C = 1.00–1.05$) with marginally higher values ($F^{14}C = 1.04–1.08$) for dissolved organic matters experimentally produced by shaking those samples in water. In contrast, bloom samples taken from a private harbor (INRAE; SP1 in Fig. 1) and from other sampling locations (SP3-SP5 and SHL2 in Fig. 1) between September 10 to 15, as well as pelagic water samples collected during October 2021, were strongly depleted ($F^{14}C = 0.806–0.884$), as was dissolved inorganic carbon at all depths ($F^{14}C = 0.851–0.860$). These contrasting results demonstrate that terrestrial organic matter initially fed this mixotroph species in the littoral, but lake bicarbonate sustained it on its blooming way to the pelagic zone.

Our results indicate that the bloom mostly originated from the southern shore and its spatial distribution was shaped by gyre-induced advection to the pelagic regions. However, freshwater primary production depends on various factors such as nutrients, light availability and suitable background conditions[26,27]. In the following, we discuss how a specific sequence of meteorological forcings set the stage for the emergence and growth of a widespread surface bloom in this stratified meso-oligotrophic lake.

## Calm and warm conditions: Creating the ideal niche for Uroglena bloom growth and dispersion

Several windless days with relatively warm weather and adequate solar energy are essential to form and maintain Uroglena colonies in the water surface layer (top 5 m)[10,28]. Meteorological records confirm the presence of such conditions during the observed bloom in Lake Geneva (red region in Fig. 3). The daily cumulative global radiation remained high during the bloom (Fig. 3c) due to the absence of clouds. Air temperature showed a drop during strong wind days (August 24–31) and positive trends in the calm period until September 8 (Fig. 3d). We observed an average net heat gain of 94 W m$^{-2}$ over this period, which is crucial for boosting phytoplankton

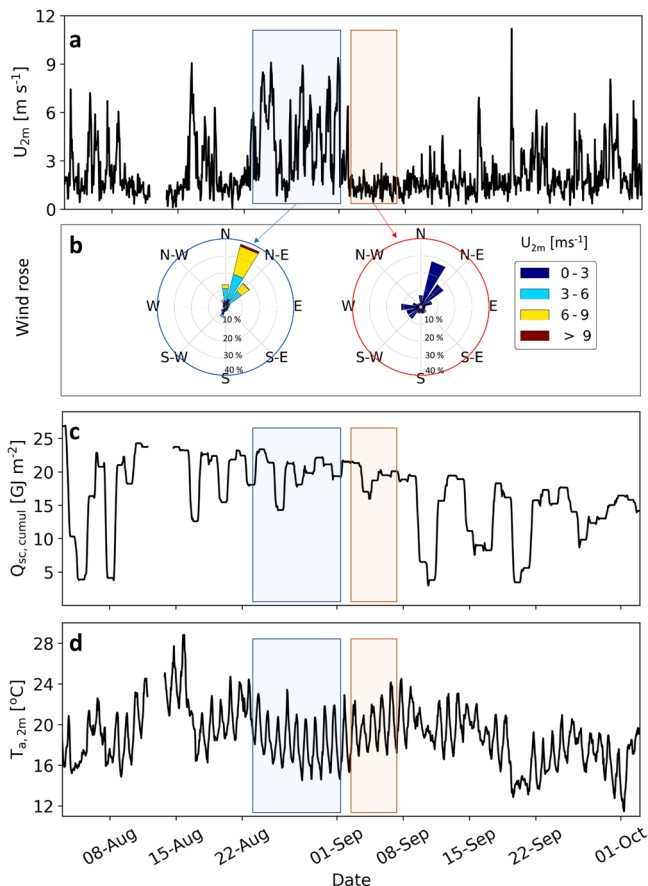

**Fig. 3 | Meteorological parameters measured at the LéXPLORE platform. a** Wind speed, **b** wind rose diagram for two selected periods in **a**, **c** global radiation converted to cumulative daily values, and **d** air temperature. The measurements show a windy cold period (blue shaded boxes) followed by a calm warm period (red shaded boxes) prior to the bloom's peak. Similar temporal patterns were obtained using the spatiotemporal reanalysis datasets over the lake from the COSMO numerical weather model (Supplementary Fig. 13).

productivity. The absence of strong surface mixing also grants the spatially heterogeneous dispersion of the bloom from littoral toward the pelagic zone through wind-induced multi-gyre circulation[29] (Fig. 2a, b; for further details, see Supplementary Note 1).

## Wind-induced coastal upwelling: A littoral nutrients fertilizer

Nutrients, which are essential for bloom development, are usually more abundant in deep than euphotic layers of low-nutrient lakes during summer stratification[26,30]. Indeed, the historical decrease of nutrient input to Lake Geneva determined a progressive deepening of the layer where phosphorus concentrations would be high enough to sustain phytoplankton growth[31]. Contrary to eutrophic lakes where nutrients concentrations do not fall below limiting concentrations that may slow down primary production, in low-nutrient lakes a strong bottom-up control is expected to prevent the bloom formation. However, upwelling of nutrient-rich hypolimnetic water to the euphotic zone is known to boost the surface layer productivity in coastal and inland waters[32–35]. Our modeling results indicated the occurrence of coastal upwelling along the southern shore a few days before the bloom onset, caused by the prevailing northeasterly winds (Fig. 3a, b and Supplementary Fig. 13E) and induced western anticyclonic gyres[24] (Fig. 2b). However, such coastal upwelling is common in Lake Geneva[36–38] and hence cannot solely explain the exceptional bloom observed in September 2021. Wind-induced upwelling events have been also reported in other large meso-oligotrophic lakes, e.g., Lake Baikal[39], Lake Tahoe[40,41], and Lake Constance[42], with some of those events leading to amplified $C_{chl-a}$[36,43,44].

## Extreme rainfall events: Excessive terrestrial organic matter and nutrients loading

In the past, Uroglena abundance has been reported in Lake Geneva only once in late July to early August 1999, mainly in the lake's south-eastern parts[45]. Climatological records of the summers of 1999 and 2021 not only share a strong wind event followed by a calm, sunny and warm period triggering the bloom, but also extreme rainfall (a 3-day cumulative value of >55 mm/day, which is >99% of the long-term record in Lake Geneva) 6–8 weeks before the blooms. The latter can explain the notable increase of lake water level (> 30 cm; Supplementary Fig. 14), and the subsequent bank erosion and induced downfalls reported in summer 2021. The high run-off provided excessive nutrient and organic matter supply through exportation of terrestrial organic matter to the lake, which illustrates our $^{14}$C dating analysis indicating a terrestrial organic carbon signature in the littoral area. Such heavy precipitation-driven pulses of terrestrial inputs have been identified as a trigger of algal bloom in lakes[3,17,46,47]. We speculate that the time-lag between the extreme rainfall event and observed Uroglena blooms was essential for incorporating terrestrial organic matter into the microbial loop favoring the growth of heterotrophic bacteria. This enabled Uroglena cells to feed on these bacteria that can be the major source of nutrients and carbon during mixotrophic growth[48] and to reach very high duplication rate[49] ultimately leading to a bloom[16,50]. Terrestrial organic carbon signature in the littoral area inferred from $^{14}$C dating analysis (Supplementary Table 2) suggests dominant mixotrophy processes rather than photosynthesis. In environments enriched in dissolved organic matter, mixotrophy is a competitive advantage for Uroglena compared to other algae species relying only on photosynthesis.

Phytoplankton control through zooplankton grazing[51,52] was also examined. The zooplankton $^{14}$C signatures were similar to the bloom signature in the open water (Supplementary Table 2) hinting that zooplankton grazing on the bloom cannot be excluded. Our measurements suggest that Daphnia might have taken advantage of the massive phytoplankton biomass but was not able to control it. This is because of the limited strength of such a top-down control in Lake Geneva[53] since herbivorous zooplankton have strongly declined following the increase in zooplanktivorous fish[54]. The limited control of Uroglena development by zooplankton could also be explained by the morphological characteristics of Uroglena, e.g., its size, colonial form and the presence of mucilage, which make it poorly vulnerable to zooplankton grazing[55].

Our results highlight the role of external precipitation-driven terrestrial source of carbon in littorals on bloom emergence in low-nutrient systems, and hence an original understanding of the high-nutrient paradigm shift[56]. Littoral zones have been acknowledged for their strong contribution in the lake productivity in small ecosystems[57,58], but, compared to pelagic zones, have been undervalued in large lakes. Recent studies indicated that littoral zones of large low-nutrient water bodies can turn into phytoplankton production hot-spots with usually higher $C_{chl-a}$ than pelagic zones and spatially heterogeneous trends[59,60].

Co-occurrence networks of bacteria and eukaryotes have already been reported in low-nutrient lakes and suggest strong interactions within the microbial community[61]. Accordingly, Chrysophytes are known to involve a variety of species exhibiting a wide range of nutritional behaviors. Among them Uroglena species are likely to outcompete other phytoplankton taxa more effectively during nutrient-stressed conditions[62]. Its capability to use bacteria as a substitutable P-source[16] offers this taxon a strong competitive advantage over other species but implies particular conditions since the presence of bacteria is mandatory for the blooming of this taxa in a laboratory culture[63].

## The importance of sequential weather events: Elucidating the rare emergence of blooms in a low-nutrient lake

Our analysis of the 2021 bloom in Lake Geneva revealed that a specific sequence of meteorologically-driven processes triggered the exceptional surface bloom in this low-nutrient system: (i) excessive organic matter and nutrients loading to the system by extreme precipitation (run-off) event, (ii) wind-induced upwelling of hypolimnetic nutrient-rich waters to the euphotic zone, followed immediately by (iii) warm (adequate temperature and solar energy) and calm (absence of bloom-breaking mixing) conditions to foster primary production in addition to mixotrophy, and thus initiate the bloom. The post-upwelling littoral-pelagic transport processes interacting with the lake surface circulation finally drew the main spatial structures observed from space, i.e., the western mushroom-like structure, the central south-to-north elongated patterns, and the eastern gyre (Fig. 1 and Supplementary Fig. 6).

To assess this hypothesis, long-term (1998–2022) meteorological data during June-September from a station around the lake (Changins in Fig. 1) were analyzed (Fig. 4a–c). The statistical results indicated only three occasions in 1999 (end-July) and 2021 (mid-July and early-September) with a significant (an arbitrary threshold of >0.75) occurrence probability of the identified sequence of meteorological conditions (Fig. 4d and Supplementary Fig. 15). The only two reported Uroglena blooms in Lake Geneva, i.e., end-July 1999 and early-September 2021, coincide with two of those occasions. The bloom absence in mid-July 2021 remains unexplained, but is likely due to the relatively short time lag between the heavy rainfall event and the occurrence of other essential meteorological conditions (<10 days in this case, compared to 6–8 weeks for the two observed blooms), which was probably insufficient for incorporating terrestrial organic matter into the microbial loop favoring Uroglena growth[63]. Our long-term analysis indicated that although each of the necessary conditions, (i) to (iii), are frequent in Lake Geneva (Supplementary Fig. 16), their exact sequence is rare. For example, the estimated bloom occurrence probability in early-June 2015, i.e., ~0.7, is also close to the arbitrary threshold. However, no bloom has been reported in the lake during this period. This can be due to at least one insignificant prerequisite, e.g., lack of warm and calm conditions in this specific example from 2015.

## Conclusions

Our study highlights the crucial role of meteorologically-driven processes and their timing in triggering Uroglena blooms in Lake Geneva, a meso-oligotrophic lake. Moreover, the results of this case-study prove that the future development of blooms do not simply follow trends in atmospheric forcing but are also closely linked to the dynamic sequence of meteorological events. For example, at our study site, more frequent and intense heavy rainfall extremes have been reported, with a projected continuation of this

**Fig. 4 | Long-term (1998–2022) measured meteorological parameters during June-September, as well as the incidents when favorable meteorological conditions for bloom occurrence were satisfied. a** 3-day cumulative precipitation, **b** estimated mean daily wind energy, **c** mean daily global radiation, and **d** calculated probability of bloom occurrence. The meteorological data are measured at Changins station (blue diamond in Fig. 1). The probability of extreme precipitation event up to 60 days before ($P_{post-precipitation}$), strong wind event a few days before ($P_{strong\ wind}$), and warm and calm conditions a few days after ($P_{warm\ \&\ calm}$) are plotted on top of panels **a**, **b**, and **c**, respectively. An arbitrary threshold of >0.75 was selected in all colormaps. The red areas in **d** indicate two occasions when Uroglena bloom were reported in Lake Geneva. Air temperature data were also used in estimating $P_{warm\ \&\ calm}$ (Eq. (13)).

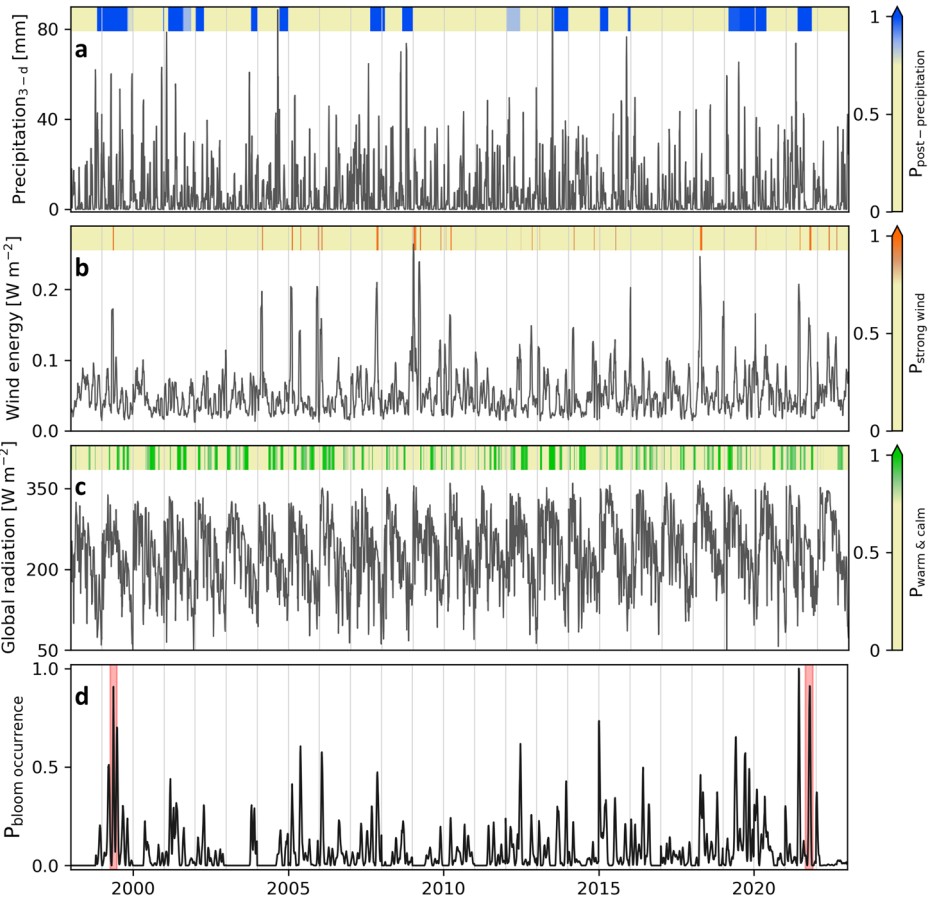

trend in a warming climate[64]. Climate models and observations also indicate an intensification of wind extremes over western and central Europe and a decrease over southern Europe. At the global scale, the intensification of extreme weather events, such as floods, droughts, and heat waves, are also expected due to climate change[65]. Such extreme events have been reported to influence the ecosystem dynamics[66,67]. However, our study suggests that the timing of favorable meteorological conditions under different climate scenarios needs to be investigated further to better understand the impact of global warming on bloom dynamics in lakes.

Our results, from a broader perspective, highlight the need to consider the lake as a dynamic ecosystem characterized by spatial heterogeneity, littoral-pelagic interconnectivities, and different communities' ecology. Our methodology and findings can be extended to other comparable aquatic systems and improve the prediction and management of algal blooms in nutrient-limited systems. The specific conditions required to trigger and sustain a specific bloom must, however, be acknowledged as they might vary depending on the ecological niches and life-traits of species. Our observations, among others[68,69], suggest mixotrophy as a selective functional trait. Mixotrophy might provide a strong advantage in oligotrophic lakes, particularly under extreme precipitation events[64,70] and subsequent water level fluctuations associated with the global warming threat. Such heavy rainfall events not only can load excessive nutrients into the water body (e.g., the massive cyanobacteria bloom in Lake Erie[17]), but also increased dissolved organic carbon inputs can be beneficial for abundance of some Chrysophytes (e.g., Uroglena blooms in Lake Geneva and Lake Beaver[3]). Nevertheless, in oligotrophic systems experiencing local phosphorus inputs, the growth of mixotrophic species can be outcompeted by toxic cyanobacteria as it has been observed in Lake Baikal[71]. Moreover, our findings underscore the need for comprehensive multidisciplinary approaches to advance our ability to understand the complex dynamics and predict the emergence of algal blooms. As next steps to improve such understanding, we suggest

incorporating growth and mortality rates into the particle tracking model, and setting up a coupled three-dimensional hydrodynamic and biogeochemical model. Hybrid deterministic and data-driven models also seem a promising alternative[72]. We finally advocate for open operational coupled Earth observation systems integrating remote sensing data, in-situ observations, and 3D models. By combining these diverse data sources, such systems enable a more accurate understanding and prediction of unexpected short-lived events, allowing for timely and informed decision-making by stakeholders.

## Methods
### Study site and monitoring data
Lake Geneva (Local name: Léman) is a large, deep perialpine lake located between Switzerland and France. It is approximately 70-km long, with a maximum width of 14 km, and a surface area of 582 km². The lake is composed of two basins: an eastern, large basin with a maximum depth of 309 m, and a western, small, narrow basin with a maximum depth of approximately 70 m. Lake Geneva has an oligomictic regime with rare complete water column mixing during exceptionally cold winters. The euphotic depth of the lake varies seasonally between ~12–25 m, and is often deeper than the mixed layer depth during spring and summer[73]. Two dominant winds called Vent and Bise, coming from the southwest and northeast, respectively, establish the lake circulation patterns. The spatio-temporal variability of meteorological forcing, combined with the Earth's rotation and lake bathymetry, gives rise to substantial hydrodynamic complexity[23,24,74].

The lake has been monitored systematically by different organizations. Bi-weekly/monthly physical and biogeochemical data have been measured by the long-term observatory and experimentation on lakes, i.e., Observatoire des LAcs (OLA) and the Commission Internationale pour la Protection des Eaux du Léman (CIPEL), at SHL2 (Fig. 1) since 1963. Lake water

level and rivers have been systematically monitored by the Swiss Federal Office of Environment (Bundesamt für Umwelt; BAFU/FOEN) since 1998. Long-term meteorological measurements are available from the station network of MeteoSwiss. In this study, the measured data at a near-shore monitoring station (Changins; Fig. 1) during 1998–2022 were used.

Complementary measurements were obtained from LéXPLORE[75] (Fig. 1), a research platform moored 570 m off the lake's northern shore. The platform is equipped with an autonomous multi-parameter profiler (WETLabs Thetis)[73]. Various bio-optical sensors and a CTD probe are mounted on this profiler. It samples the first 80 m of water column 2–8 times a day with a vertical resolution of 1–10 cm. In this study, we used CTD temperature profiles, hyperspectral absorption (from AC-S; 81 channels between 400–730 nm), backscattering at 700 nm at 117° and chlorophyll-a fluorescence (from ECO Triplet BBFL2w), as well as Photosynthetically Active Radiation (PAR; between 400 to 700 nm; from Sea-Bird ECO PARs). The high-resolution profiles were acquired at a 12-h frequency during our study period, i.e., August-September 2021. The high-resolution vertical profiles of current (below 11 m depth) were monitored by a downward-looking 300 KHz Teledyne RD Instruments. A closely-spaced thermistor chain (48 RBR temperature sensors covering 0–90 m depth range) measured stratification evolution at this platform. Additionally, high-frequency temperature data were acquired at Buchillon station (a 100-m offshore station; Fig. 1) at 1 m depth using a PT100 sensor (from Vaisala). Furthermore, a few days after the bloom's peak (on September 10 and 13) water samples were collected at five different locations around the lake (SP1-SP5 in Fig. 1) for biogeochemical analyses. Laboratory analysis of these samples are detailed below.

### In-situ data processing

In-situ $C_{chl-a}$ were calculated based on the absorption line-height ($a_{LH}$) method[76]. The $a_{LH}$ is calculated from hyperspectral absorption ($a$) measurements as:

$$a_{BL}(676\,nm) = \frac{a(715\,nm) - a(650\,nm)}{715 - 650}(676 - 650) + a(650\,nm)\,[m^{-1}] \tag{1}$$

$$a_{LH}(676\,nm) = a(676\,nm) - a_{BL}(676\,nm)\,[m^{-1}] \tag{2}$$

This technique has been reported to avoid the non-photochemical quenching effect inherent in fluorometer measurements[77]. To estimate $C_{chl-a}$, we first found the linear relationship between nighttime (low light) $a_{LH}$ and $C_{chl-a}$ measured by fluorometer for each night. The obtained relationships (with an average coefficient of determination of $R^2 = 0.94$) were then applied to daytime $a_{LH}$ to obtain $C_{chl-a}$. Subsurface chl-a maxima were estimated following Zhao et al.[78]. Since hyperspectral absorption measurements are the core of such an estimation, several steps of quality control were implemented on the AC-S data to minimize various errors. We applied a temperature correction[79] using the coefficients provided by the manufacturer and the water temperature from the CTD onboard the Thetis profiler to minimize the effects of optical misalignments and drift in AC-S readings. The M2 model from Pitarch et al.[80] was used to correct for scattering errors. Furthermore, depth profiles of absorption and attenuation spectra were investigated systematically to identify and reanalyze the errors in the form of missing values, noise, or non-physical values due to instrumental artifacts (e.g., random electronic noise) or poor sample baselines. Processing and quality control of AC-S data are detailed in the supporting information of Minaudo et al.[73].

$bb_{700nm}$ measurements, providing a proxy for the suspended particles concentration[81], were also used. The backscattering profiles were corrected for instrument drift using pre- and post-deployment manufacturer calibration factors. A linear drift in time was assumed and the relationship between backscattering coefficient and volume scattering function at the measurement angle (117°) was implemented for this correction[82].

Euphotic depth and $z_{SD}$, representing light availability and water column transparency, were estimated based on PAR measurements. The euphotic depth is usually defined as the depth where PAR values attain 1% of the surface values[83]. To compute $z_{SD}$, the diffuse attenuation coefficient of PAR ($k_{d,PAR}$) was first estimated by fitting an exponential curve to the top 10 m of PAR measurements[84]:

$$PAR(z) = PAR(z_{0-})e^{-k_{d,PAR}z} \tag{3}$$

where $PAR(z_{0-})$ is the subsurface PAR value. Secchi depth was then estimated as[85,86]:

$$z_{SD} = \frac{1.44}{k_{d,PAR}} \tag{4}$$

Thermocline depth was calculated based on statistical segmentation of temperature profiles into three layers, using the density gradient values at each depth, and defining the thermocline as the point in the middle layer where the absolute value of the density gradient is a maximum[87].

### Satellite data retrievals

$R_{rs}$ products from Sentinel-2 MSI and Sentinel-3 OLCI were retrieved using ACOLITE atmospheric correction[88]. The Three-Band reflectance Index (TBI), a proxy for $C_{chl-a}$, was calculated from remote sensing reflectance products at three red and near-infrared wavelengths (665, 704, and 740 nm for Sentinel-2 MSI, and 665, 709, and 754 nm for Sentinel-3 OLCI)[15,89] as:

$$TBI = \left(\frac{1}{R_{rs}(665nm)} - \frac{1}{R_{rs}(704\,or\,709nm)}\right)R_{rs}(740\,or\,754nm) \tag{5}$$

Several bio-optical models[90] were tested to estimate $C_{chl-a}$, where the following returned the most reasonable $C_{chl-a}$ range (Supplementary Fig. 6A) as expected for the bloom event:

$$C_{chl-a} = 98.77(TBI) + 34.76 \tag{6}$$

We used RGB-QAA algorithm[91] to obtain $z_{SD}$ from the Sentinel-2 MSI image on 6 September, 2021 (Supplementary Fig. 6B).

### Water samples analyses

Water samples were taken from Lake Geneva at different depths (close to the surface, the depth of maximum $C_{chl-a}$, and/or metalimnion) on September 10 (SP1-SP5 in Fig. 1) and September 13 (SP1-SP2 in Fig. 1). Species identification and counting were performed in sedimentation chambers under an inverted microscope[92]. On September 13, $z_{SD}$ and $C_{chl-a}$ measurements with a multiparametric probe were performed at three points on a littoral-pelagic transect at SP5. Measured $C_{chl-a}$ were not corrected for quenching, and thus might be underestimated. Additional bulk phytoplankton and water sampling have been performed at SHL2 (Fig. 1) during the lake monitoring for $^{14}C$.

Soil and sediment were sampled with a spade and plastic bags, taking four samples: (i) a silty surface (0–10 cm) sediment from the side of the Redon river (close to SP2 in Fig. 1), 50 m from the lake, (ii) the soil (0–10 cm) from an artificial lawn used as a beach near the Redon mouth, (iii) a subsurface (depth 30–50 cm) silty-clay horizon from the eroded littoral path around the lake, and (iv) a sandy gravel littoral accumulation (0–10 cm) at the foot of a steep slope, close to Thonon-les-Bains. Samples were dried at 35 °C and sieved (2 mm).

Soil organic matter is enriched in nuclear bomb $^{14}C$ ($F^{14}C > 1.00$). In contrast, lake dissolved inorganic carbon is depleted ($F^{14}C < 1.00$) because the lake watershed is mainly based on limestone rocks which do not contain $^{14}C$. For $^{14}C$ dating, samples were handled by the ECHoMICADAS facility at LSCE[93]. Chemical protocol was applied according to the sample specificity[94], and measurement performed on either the solid or the gas source,

depending on the amount of carbon available in the sub-sample made in the lab. The following procedure was implemented:

(i) Soil and sediment: $CaCO_3$ was removed by successive additions of 10 μL of 1 N HCl, under binocular magnifier, in the EA capsule. Then capsules were freeze-dried, closed, and introduced in the ECHoMI-CADAS gas source through the EA-GIS interface[59]. In addition, to simulate the action of waves, soil samples (40 g) were shaked (200 rpm) in distilled water (200 mL) for 7 days at 25 °C in the dark and dissolved organic carbon was processed as below.

(ii) Dissolved inorganic carbon was extracted from water (previously filtered at 0.7 μm). Under helium flow, the water was acidified, releasing the dissolved inorganic carbon as $CO_2$ which was dried and trapped, and sent to the national facility, LMC14, for reduction and measurement[95].

(iii) Dissolved organic carbon was retrieved from water filtered at 0.7 μm and frozen-dried. $CaCO_3$ removal and measurements were performed as for soil.

(iv) Particulate organic matter was retrieved on pre-combusted 0.7 μm glass fiber filter (GF/F) by filtering water and rinsed to remove $CaCO_3$. A punch was taken to evaluate the carbon content and thus the size of the punch to be taken for a $^{14}C$ measurement on a gas source through the EA-GIS interface (50–100 μgC). The measurements were duplicated to estimate the heterogeneity of the sample.

(v) Crustacean samples were acid-washed (HCl 1.2 N), then transformed into $CO_2$ and reduced to C graphite with the Automatic Graphitization Equipment (AGE3)[96].

## 3D numerical model

A 3D numerical model based on a hydrostatic version of the MITgcm code[20] was employed to simulate the lake hydrodynamics. The model configurations were adopted from Safin et al.[20]. They calibrated different model parameters (e.g., Smagorinsky viscosity and Dalton number) for Lake Geneva based on current velocity and temperature measurements at one and four stations, respectively. Their analysis during 15 January–15 December 2019 indicated a temperature root mean square error (RMSE) of 0.95–1.45 [°C] across different monitoring stations, and a velocity RMSE of 0.033 [m s$^{-1}$] compared to the measurements at LéXPLORE station (Fig. 1). The simulations in this study were performed on a Cartesian grid with 50 m horizontal resolution and 100 vertical layers increasing gradually from 0.5 m at the surface to 10.1 m at the lake's deepest point (i.e., SHL2 in Fig. 1). The model was initialized from rest with horizontally homogeneous temperature profiles on July 26, 2021 obtained from measurements at the LéXPLORE station (red square in Fig. 1). This particular day was selected because the preceding days were characterized by calm wind conditions. Meteorological reanalysis data over the lake from the COSMO-1 numerical weather model[97] provided by MeteoSwiss on a 1.1 km × 1.1 km was used to force the model. Monthly-averaged Secchi depth measurements were used for modeling solar energy penetration into the water. The comparison of the model results with observations indicated a spin up time of ~3 weeks (Supplementary Fig. 1). However, the results obtained 35 d after the model initialization were mainly used in our study (Period of interest in Supplementary Fig. 1B). Model validation and evaluation during our studied period are discussed in Supplementary Methods and supported by Supplementary Figs. 1–5.

## Lagrangian particle tracking

The Lagrangian particle tracking code employed here is based on the method proposed by Döös et al.[21], which has been adopted and validated for Lake Geneva[37,38,98–100]. This method tracks passively the 3D Lagrangian motion of inert particles with no mass and no volume using the velocity field obtained from the hydrodynamic model. In our Lagrangian simulations, diffusion (e.g., by random particle displacement) was neglected and particle trajectories were followed based on 3D advection solely. Given the bloom's visually distinguishable spatial patterns (Fig. 1), the advection of water masses was indeed assumed as the dominant process in its dispersion. In order to reproduce those large-scale observed patterns, we followed the

deterministic advection by the flow resolved in the hydrodynamic model while not being in smaller-scale turbulent mixing.

Based on satellite imagery, we assumed that the onset of the bloom took place mainly along the southern shores (due to wind-induced coastal upwelling shown in Fig. 2c, d, which presumably brings external C inputs to the euphotic zone), starting around September 3 (see satellite images in Supplementary Fig. 10). Building on this knowledge, 2000 particles were released every hour in the near-surface layers (0–10 m depth) of the upwelling regions along the southern shore, starting on September 3, 2021 at 00:30. To account for uncertainties in the flow field, the exact seeding locations were selected randomly at each hourly release. Furthermore, three backward tracking simulations were performed. Here, 200,000 particles were instantaneously released on September 6, 2021 at 12:30 at the centers of the three main circulation patterns in the near-surface layer (0–5 m), and were tracked backward until August 31, 2021 at 00:30.

For the forward tracking simulation, the relative particle concentration at each model grid cell was computed. To better compare the results with the satellite observations, a depth weighting function was implemented following Piskozub et al.[101] to compute the depth-integrated concentration seen by satellite sensors from the simulated depth layers at each time step. A constant diffuse attenuation coefficient was assumed, which was estimated to be ~0.41 m$^{-1}$ based on in-water PAR measurements (Eq. (3)) during September 3–6. For the backward tracking simulations, the particle origins were traced by analyzing particle probability maps. The latter were computed as the total number of particles that visited each grid cell during the entire simulation divided by the total number of particles[100].

## Empirical Orthogonal Function (EOF) analysis

To identify the main coherent spatial patterns of the horizontal velocity field, an EOF analysis[102] was carried out. Here, the horizontal velocity flux (φ), i.e., the vertically integrated horizontal velocity ($u_h$) was used:

$$\vec{\phi} = \int_{z_1}^{z_2} \vec{u_h}\,dz\,dr \tag{7}$$

Velocity flux φ is usually a more relevant quantity than the vertically averaged velocity for investigating the basin-scale transport in large lakes[99].

EOF analyses decompose the spatio-temporal variability of a geophysical field into a linear combination of orthogonal spatial patterns (E) that evolve temporally (T):

$$\vec{\phi}(\vec{x}, t) = \sum_{k=1}^{N} \vec{E}_k(\vec{x}) T_k(t) \tag{8}$$

where $\vec{\phi}(\vec{x}, t)$ is the target geophysical flow field (horizontal velocity flux in our case), and $k$ is the EOF mode at location $\vec{x}$ and time t. If a dataset contains coherent structures, they are usually detectable with the first few EOF modes together with their corresponding principal component time series. Here, we identified the main spatial mode of the horizontal velocity flux field in the surface layer using the modeled depth-averaged (0–20 m depth) horizontal velocity fields. The results indicated a three-gyre structure in the lake (Fig. 2b). Averaging the velocity fluxes between 0–10 m depth did not change the results significantly.

## Long-term statistical analysis of meteorological data

Decadal meteorological data are available from a few on-shore stations surrounding Lake Geneva. Here, we used daily mean wind speed ($U_{10}$), global radiation ($Q_{sc}$), air temperature ($T_a$), and 3-day cumulative precipitation (pr) at Changins station (blue diamond in Fig. 1) to assess the occurrence of the required sequence of meteorological forcing. To estimate the effect of wind speed on lake mixing and transport, the wind

surface energy flux ($P_{10}$ in W m$^{-2}$) was calculated[103] as:

$$P_{10} = \rho_{air} C_{10} U_{10}^3 \qquad (9)$$

where $\rho_{air}$ is the air density (assumed a constant value of 1.24 kg m$^{-3}$) and $C_{10}$ is the wind drag coefficient estimated as a function of wind speed at 10 m ($U_{10}$)[104].

High-frequency variations of each measured or estimated parameter were first removed by applying a 3-day moving average. Then, we mapped the data linearly to a standard range using predefined lower/upper boundaries (indicated below), i.e., assigning 0 and 1 to lower and upper boundaries, respectively. Finally, to smooth the mapping and avoid any values out of 0 to 1 range, a 3$^{rd}$ order normalized sigmoid function was implemented:

$$S(y_i^n) = 1 - \frac{2}{1 + e^{3y_i^n}}, 0 \le y_i^n \le 1 \qquad (10)$$

where S is the sigmoid function, and $y_i^n$ is the linearly mapped dataset $y_i$ ($U_{10}$, $Q_{sc}$, $T_a$, or pr) at time n.

The 5 and 95 percentiles of summertime (June to September) surface wind energy data, corresponding to 0.03 and 0.12 W m$^{-2}$, respectively, were assumed as the mapping extreme values for wind speed measurements. A higher threshold of 99 percentile was assumed for the precipitation data. This is equivalent to 55 mm of 3-day cumulative rainfall. To detect heavy rainfall events, the extreme linear mapping values were considered with a small variation around it, i.e., 50 mm and 60 mm. For global radiation and air temperature, however, more relaxed values were employed for the mapping. We used 1 and 50 percentile of summertime data for the lower and upper boundary values, respectively. These values are 81 and 152 W m$^{-2}$ for $Q_{sc}$, and 10 and 18 °C for $T_a$.

Then, at each time step, we calculated the probability of each of the three identified prerequisites for an algal bloom, i.e., heavy rainfall a few weeks before, strong wind energy a few days before, and calm and relatively warm conditions a few days after (Fig. 4 and Supplementary Fig. 15).

$$P_{post-precipitation}^n = P_1^n = \max\left[S\left(y_{Pr}^j\right)\right], j \in \{n-59, n-58, \ldots, n\} \qquad (11)$$

$$P_{strong\ wind}^n = P_2^n = mean\left[S\left(y_{U10}^j\right)\right], j \in \{n-6, n-5, \ldots, n\} \qquad (12)$$

$$P_{calm\ \&\ warm}^n = P_3^n = mean\left[S\left(y_{Ta}^j\right)\right] \times mean\left[S\left(y_{Qsc}^j\right)\right]$$
$$\times mean\left[1 - S\left(y_{U10}^j\right)\right], j \in \{n, n+1, \ldots, n+6\} \qquad (13)$$

Here, the time lags (for "post-precipitation" and "strong wind") and time lead (for "calm & warm") of 60 days, 7 days, and 7 days, were assumed for the three prerequisites, respectively, indicated by j in Eqs. (11)–(13). These time lags and time lead are slightly higher than those found for 2021 Uroglena bloom in Lake Geneva, i.e., ~6 weeks for heavy rainfall, ~4 days strong wind, and ~4 days for calm & warm conditions. This will give more flexibility to the statistical model, i.e., more days will be attributed with favorable prerequisites. Finally, the bloom occurrence conditional probability was estimated as follows:

$$P^n(bloom) = \prod_{i=1}^{3} P_i^n \qquad (14)$$

An arbitrary threshold of >0.75 was employed as the significant probability of bloom occurrence. This relatively high value corresponds to a probability of at least 0.9 for each of three prerequisites. However, by mathematical definition, the lower the value of $P_n$ (bloom), the less chance there is for the bloom occurrence.

## Reporting summary

Further information on research design is available in the Nature Portfolio Reporting Summary linked to this article.

## Data availability

The LéXPLORE and Buchillon datasets are openly available at www. datalakes-eawag.ch. The source data for regenerating the graphs and plots in the manuscript are available for download at https://doi.org/10.25678/000C9Q. Long-term systematic water quality measurements at SHL2 were obtained from https://si-ola.inrae.fr/si_lacs/login.jsf. Reanalysis surface forcing COSMO data can be requested from MeteoSwiss (https://www.meteoswiss.admin.ch/). Water level data can be demanded in the "Hydro-logical Data Service" section of BAFU/FOEN website: https://www.bafu.admin.ch/.

## Code availability

The main hydrodynamic code is available at a modified MITgcm repository[20]: https://renkulab.io/projects/artur.safin/DatalakesHydrodynamics. Lagrangian C-tracker particle tracking code is available at https://zenodo.org/record/1034118.

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

## Acknowledgements
We thank the entire *LéXPLORE* consortium for the administrative and technical support and especially the five involved partner institutions: University of Lausanne, EPFL, Eawag, University of Geneva, and INRAE-USMB (CARRTEL). We especially thank Sébastien Lavanchy and Guillaume Cunillera for their technical support. We also wish to thank James Runnalls for developing the sensor-to-frontend data pipeline providing near-real-time visualization and downloading of *LéXPLORE* and *Buchillon* core dataset. Authors gratefully acknowledge the OLA observatory, Raphaël Jordan and Fabio dos Santos Correia for the samples acquisition during the bloom, and Nina Autant for the estimation of *Uroglena* abundances. We also wish to thank Artur Safin and Cintia Ramón Casañas for their helpful discussion on 3D hydrodynamic modeling. Long-term in-situ data at SHL2 location in Lake Geneva were collected and stored with support from the © OLA-IS, AnaEE-France, INRAE of Thonon-les-Bains, CIPEL[105]. We thank the Swiss Federal Office of Meteorology and Climatology, MeteoSwiss, for providing the spatiotemporal meteorological data. We would like to thank the Swiss Federal Office for the Environment (BAFU/FOEN) for water level data. We are also thankful to Jennifer Susan Adams for her valuable insights on the textual aspects of this manuscript, and to Rosi Silber for providing us with the topographic features in Fig. 1. This work was supported by ESA through the grant AlpLakes (Contract No.4000136401; AO/1-8216/15/I-SBo; Scientific Exploitation of Operational Missions - SEOM S2_4SCI Land and Water Coastal and Inland Waters), and by the Swiss National Science Foundation grant Lake3P, no. 204783. Financial support for isotopes analysis was provided by CIPEL, INRAE and University of Savoie Mont-Blanc. We thank the anonymous reviewers for their constructive comments and suggestions that helped improve the paper.

## Author contributions
Author credit reflects the independent view of the author.Conceptualization: A.I.R., D.B., D.O., E.D., T.H., S.J. and O.A. Methodology: A.I.R., D.O., T.H., O.A., E.D., C.H., D.J., F.R., V.T.-K. and D.B. Software: A.I.R., D.O., R.S.R. and M.W. Validation: A.I.R. Formal analysis: A.I.R., C.H., F.R. and V.T.-K Investigation: A.I.R., D.O., O.A., O.S.S., R.S.R., M.A., M.T., S.J., T.H., M.W., F.S., E.D., D.J., C.H., V.T.-K., S.R., F.R. and D.B. Resources: A.I.R., O.A., E.D., F.S. and V.T.-K. Visualization: A.I.R. Supervision: D.B., D.O., O.A. and E.D. Writing—original draft: A.I.R. Writing—review and editing: A.I.R., D.O., O.A., O.S.S., R.S.R., M.A., M.T., S.J., T.H., M.W., F.S., E.D., D.J., C.H., V.T.-K., S.R., F.R. and D.B. Funding acquisition: D.B., O.A., E.D. and F.S. All authors have read and approved the final version of the manuscript.

## Competing interests
The authors declare no competing interests.

## Additional information

[1]Eawag, Swiss Federal Institute of Aquatic Science & Technology, Surface Waters – Research and Management, 8600 Duebendorf, Switzerland. [2]Department of Geography, University of Zurich, 8057 Zurich, Switzerland. [3]Université Savoie Mont Blanc, INRAE, UMR CARRTEL, 74200 Thonon-les-Bains, France. [4]Eawag, Swiss Federal Institute of Aquatic Science & Technology, Surface Waters – Research and Management, 6047 Kastanienbaum, Switzerland. [5]Department of Civil & Environmental Engineering, University of California, Davis, Davis, CA, USA. [6]Ecological Engineering Laboratory (ECOL), Institute of Environmental Engineering (IIE), Faculty of Architecture, Civil and Environmental Engineering (ENAC), Ecole Polytechnique Fédérale de Lausanne (EPFL), 1015 Lausanne, Switzerland. [7]Institute for

Electromagnetic Sensing of the Environment (IREA), National Research Council of Italy (CNR), 20133 Milan, Italy. [8]Department of Civil, Environmental and Mechanical Engineering, University of Trento, 38122 Trento, Italy. [9]Earth Observation Unit, Magellium, Toulouse, France. [10]Commission Internationale pour la Protection des Eaux du Léman (CIPEL), Nyon, Switzerland. [11]Université Paris Cité, Institut de Physique du Globe de Paris, CNRS, 75005 Paris, France. [12]Laboratoire des Sciences du Climat et de l'Environnement, CEA, CNRS, UVSQ, Université Paris-Saclay, 91191 Gif-sur-Yvette, France. [13]Institute of Physics, Silesian University of Technology, 44-100 Gliwce, Poland. [14]Faculty of Geosciences and Environment, Institute of Earth Surface Dynamics, University of Lausanne, Geopolis, Mouline, 1015 Lausanne, Switzerland. ✉e-mail: abolfazl.irani@eawag.ch

