## [Peer Review File · Communications Earth & Environment]

7th Nov 23

Dear Dr Irani Rahaghi,

Your manuscript titled "Combined Earth observations unravel the complex dynamics of a freshwater algal bloom" has now been seen by 3 reviewers, whose comments are appended below. You will see that they find your work of some potential interest. However, they have raised quite substantial concerns that must be addressed. In light of these comments, we cannot accept the manuscript for publication, but would be interested in considering a revised version that fully addresses these serious concerns.

We hope you will find the reviewers' comments useful as you decide how to proceed. Should additional work allow you to address these criticisms, we would be happy to look at a substantially revised manuscript. If you choose to take up this option, please either highlight all changes in the manuscript text file, or provide a list of the changes to the manuscript with your responses to the reviewers. In particular, please consider the following editorial thresholds: 1) please provide in-deep discussion the role of biological processes like competition and grazing in the dynamics of harmful algal blooms, 2) provide quantitative analysis of hydrodynamic and biogeochemical variables affecting the model, and 3) include validation and evaluation analyses of the model.

If the revision process takes significantly longer than three months, we will be happy to reconsider your paper at a later date, as long as nothing similar has been accepted for publication at Communications Earth & Environment or published elsewhere in the meantime.

Please use the following link to submit your revised manuscript, point-by-point response to the reviewers' comments with a list of your changes to the manuscript text (which should be in a separate document to any cover letter), a tracked-changes version of the manuscript (as a PDF file) and any completed checklist:

[link redacted]

Please do not hesitate to contact us if you have any questions or would like to discuss the required revisions further. Thank you for the opportunity to review your work.

Best regards,

Jose Luis Iriarte Machuca, PhD
Editorial Board Member
Communications Earth & Environment

Clare Davis, PhD
Senior Editor
Communications Earth & Environment

EDITORIAL POLICIES AND FORMAT

If you decide to resubmit your paper, please ensure that your manuscript complies with our editorial policies and complete and upload the checklist below as a Related Manuscript file type with the revised article:

Editorial Policy Policy requirements (Download the link to your computer as a PDF.)

For your information, you can find some guidance regarding format requirements summarized on the following checklist: (<https://www.nature.com/documents/commsj-phys-style-formatting-checklist-article.pdf>) and formatting guide (<https://www.nature.com/documents/commsj-phys-style-formatting-guide-accept.pdf>).

REVIEWER COMMENTS:

Reviewer #1 (Remarks to the Author):

Review of “Combined Earth observations unravel the complex dynamics of a freshwater algal bloom” by Irani Rahaghia et al.

This article discusses a rare and large *Uroglena* phytoplankton bloom in Lake Geneva in September 2021. The authors argue that the conditions leading to the formation of this bloom are explained by a relatively rare sequence of meteorological events: a period of heavy rainfall, a period of upwelling favorable winds, followed by a period of warm and calm conditions to reduce vertical mixing. The authors go on to demonstrate that 2 of the 3 massive *Uroglena* blooms observed in Lake Geneva in recent history occurred following these same conditions. The main conclusion is that a combination of biogeochemical analysis, hydrodynamic modelling, satellite remote sensing, and in situ measurements leads to a powerful multi-disciplinary approach for understanding and explaining the cause and evolution of rare phytoplankton blooms. In other words, the factors that lead to rare bloom formation extend beyond a balance between nutrient loading and increasing temperatures which is often the focus in the context of a changing climate.

I found the article to be interesting, relevant and clearly written, however, I feel that there was still some room for improvement. While I appreciated the thorough exploration of the physical and environmental factors associated with this bloom, I was also left wondering about other biological factors, such as predation and competition, which might be relevant in this rare bloom occurrence. As such, I recommend a revision and would reconsider my position after the following comments are addressed either through a rebuttal or modifications to the manuscript.

General comments

1. The description of the Uroglena phytoplankton could be expanded and improved upon which would aid in understanding additional factors that might impact the evolution of the bloom. For example, are there any predation or grazing pressures on this particular species? Other studies have indicated a strong link between “top-down” controls and phytoplankton biomass (e.g. Anneville et al, 2019; Tadonl  k   et al. 2009). Are those pressures not relevant for this particular species of phytoplankton and this bloom event? If so, I think it would be helpful to explain that so that readers can determine how these results could inform bloom dynamics for other species and regions.
2. The qualitative and visual match between the particle tracking results in Figure 2A and the satellite observations of the bloom in Figure 1 are quite impressive. In addition, I appreciate the hydrodynamic explanation of the large geographical extent of the bloom. However, I’m having a hard time assessing whether or not these results are physically consistent. Part of the concern is that there is no assessment of the 3D hydrodynamic model validity and the details of the model configuration are lacking. For example, the model stratification and currents are not compared with observations. In addition, there is not a description of the advective time scales that might be expected during the simulation period. Is it possible to add an assessment of the model currents and stratification, perhaps at the L  XPLORE station, to the Supplementary Information? That addition would give some confidence in the model’s ability to predict the advection and dispersion of the bloom, beyond a qualitative, graphical comparison, especially where there is some uncertainty with respect to the start date of the bloom.
3. Related to the comment above, the 3D hydrodynamic model is initialized from rest with a horizontally uniform temperature field approximately 1-month prior to the time period of interest. Is a one-month spin-up period adequate? Does this give enough time for the temperature field and currents to adjust from the unphysical initial conditions? For example Cimatoribus et al (2018) used a spin up period of over 3 months. More explanation is needed to justify the relatively short spin up period.
4. Are the Lagrangian particles subject to any vertical motions (e.g. vertical mixing, vertical velocities, buoyancy or sinking)? Or are they advected horizontally by the two-dimensional model currents at the appropriate depth level?

Specific comments

1. Figure 2B: Do the arrows have units? If so, what are they? If not, I’m confused about how the EOF decomposition of the velocity field described in equation 7 relates to this figure given the units of the magnitude time series are m^3/s .
2. Figure 4C: I’m confused by the label “global radiation”. Is this really a global radiation time series? Wouldn’t it be more relevant to show the local solar radiation conditions?
3. Page 13, first paragraph: What does OLA stand for?

References

- Anneville, O., Chang, C.-W., Dur, G., Souissi, S., Rimet, F. and Hsieh, C.-h. (2019), The paradox of re-oligotrophication: the role of bottom-up versus top-down controls on the phytoplankton community. *Oikos*, 128: 1666-1677. <https://doi.org/10.1111/oik.06399>
- R  my D. Tadonl  k  , J  r  me Lazzarotto, Orlane Anneville, Jean-Claude Druart, Phytoplankton productivity increased in Lake Geneva despite phosphorus loading reduction, *Journal of Plankton Research*, Volume 31, Issue 10, October 2009, Pages 1179–1194, <https://doi.org/10.1093/plankt/fbp063>
- Cimatoribus, A. A., Lemmin, U., Bouffard, D., & Barry, D. A. (2018). Nonlinear dynamics of the

nearshore boundary layer of a large lake (Lake Geneva). *Journal of Geophysical Research: Oceans*, 123, 1016–1031. <https://doi.org/10.1002/2017JC013531>

Reviewer #2 (Remarks to the Author):

This study focuses on the dynamics of *Uroglena* Sp. algal blooms in the low-nutrient Lake Geneva. It puts forward an interesting sequence of events that can trigger the occurrence of such blooms, which, given the low nutrient concentration, in the lake are quite rare. The sequence involves heavy rainfall and wind-induced coastal upwelling to increase the nutrient concentration in the lake's top layer and hence trigger the bloom followed by a prolonged period of warm and calm weather to sustain the bloom's development. The authors nicely combine in-situ measurements, satellite remote sensing and hydrodynamic modelling to make their point.

Although I am not an expert in freshwater algal blooms and thus unable to thoroughly evaluate the novelty of this study, I found it to be well-constructed, clear, and compelling. I would therefore recommend its publication.

My main concerns/recommendations are the following:

1. While the sequence of events that the authors have identified can explain the occurrence of algal blooms in end-July 1999 and early-Sept. 2021, it is not clear how often the same sequence of events was observed in Lake Geneva *without* triggering and sustaining a bloom. I think the authors should better discuss whether such « false positive » events occurred. On page 10, they mention the absence of a bloom in mid-July 2021 without providing much details about it. Were there other similar events?

2. I also regret the lack of generalization to other oligotrophic lakes (like e.g. Lake Baikal or Lake Tahoe). Is there any indication that the conclusions of this study could be applied elsewhere or have indeed been observed elsewhere? I think that by broadening the scope of their study beyond Lake Geneva, the authors would make it stronger.

3. I also regret that the authors didn't go one step further in their modelling work of the lake by considering a biogeochemical model and assessing whether such a model produced a bloom when the sequence of events put forward by the authors is satisfied. I understand that models remain simplifications of reality that don't capture all the processes driving the lake's dynamics. They remain however very useful to test assumptions. By running such a biogeochemical model, the authors would have obtained a more quantitative understanding of the importance of each step in the sequence they put forward and be able to assess e.g. what amount nutrient input is needed to trigger the bloom and hence how long the coastal upwelling should be sustained to achieve that.

A more minor comment concerns the description of the Lagrangian particle tracking (LPT) model where the authors indicate that they neglect diffusion (p.17). That seems quite unrealistic as, even if advection of the bloom by the currents is main driving mechanism, the lake remains turbulent and diffusion is hence present. I'm wondering whether the authors are not making this assumptions because they are running the LPT model backward in time and are hence uncomfortable with the idea of reversing the sign of diffusion, which is of course unphysical. I would advise them to have a look at the following paper that nicely explains how to account for diffusion in backward simulations:

<http://journals.ametsoc.org/doi/10.1175/JTECH1874.1>

Reviewer #3 (Remarks to the Author):

This manuscript presents a combined earth observation approach, including satellite remote sensing, 3d-hydrodynamic modeling, in-lake water quality and optical observation, and statistical modeling of meteorological data to explain the occurrence of an algal bloom event in Lake Geneva, Switzerland/France. The authors present sound methods for evaluating different components of the lacustrine processes contributing to the bloom event. The manuscript is very well-written and well-thought-out. However, part of the discussion reads like it belongs in the Results section (e.g., the information on the three sequences of meteorological forcing).

My only concern regarding this manuscript is the generalizability of the core findings. Meteorological forcing is the only novel contribution in this manuscript that has not been robustly validated. The probability of bloom occurrence numbers need to be investigated further. What does a probability value less than the arbitrary value of 0.75 mean? Is this approach lake-specific, or can it work elsewhere? The authors claim these methodologies and findings can be extended to comparable aquatic systems. However, it has yet to be demonstrated in a lake other than Lake Geneva. Can the meteorological patterns in another lake with similar trophic levels detect the occurrence of algal bloom events? Without an investigation like that, the findings presented here would be a case study.

A minor note:

The remote sensing method adopted here is often termed as a three-band algorithm. The algorithm does not represent the NIR peak (the term used here). Instead, the algorithm semi-analytically measures an optical proxy of chl-a absorption at 665 nm.

Response to reviewers

Manuscript ID: COMMSENV-23-1428-T

Original title: *“Combined Earth observations unravel the complex dynamics of a freshwater algal bloom”*

Revised title: *“Combined Earth observations illuminate the path to a freshwater algal bloom emergence”*

Authors: *Abolfazl Irani Rahaghi, Daniel Odermatt, Orlane Anneville, Oscar Sepúlveda Steiner, Rafael Sebastian Reiss, Marina Amadori, Marco Toffolon, Stéphan Jacquet, Tristan Harmel, Mortimer Werther, Frédéric Soullignac, Etienne Dambrine, Didier Jézéquel, Christine Hatté, Viet Tran-Khac, Serena Rasconi, Frédéric Rimet, Damien Bouffard*

This document follows the following fonts and colors:

- *Reviewer comment*
- Authors response
- **Quote from the revised manuscript**
- *L#: Line numbers in the revised manuscript without track-changes*

Reviewer #1

The authors go on to demonstrate that 2 of the 3 massive Uroglena blooms observed in Lake Geneva in recent history occurred following these same conditions. The main conclusion is that a combination of biogeochemical analysis, hydrodynamic modelling, satellite remote sensing, and in situ measurements leads to a powerful multi-disciplinary approach for understanding and explaining the cause and evolution of rare phytoplankton blooms. In other words, the factors that lead to rare bloom formation extend beyond a balance between nutrient loading and increasing temperatures which is often the focus in the context of a changing climate. I found the article to be interesting, relevant and clearly written, however, I feel that there was still some room for improvement. While I appreciated the thorough exploration of the physical and environmental factors associated with this bloom, I was also left wondering about other biological factors, such as predation and competition, which might be relevant in this rare bloom occurrence. As such, I recommend a revision and would reconsider my position after the following comments are addressed either through a rebuttal or modifications to the manuscript.

We are thankful for the Reviewer's comments. Please see our responses and modifications to each of those concerns below.

General comments

1. The description of the Uroglena phytoplankton could be expanded and improved upon which would aid in understanding additional factors that might impact the evolution of the bloom. For example, are there any predation or grazing pressures on this particular species? Other studies have indicated a strong link between "top-down" controls and phytoplankton biomass (e.g. Anneville et al, 2019; Tadonléké et al. 2009). Are those pressures not relevant for this particular species of phytoplankton and this bloom event? If so, I think it would be helpful to explain that so that readers can determine how these results could inform bloom dynamics for other species and regions.

*Anneville, O., Chang, C.-W., Dur, G., Souissi, S., Rimet, F. and Hsieh, C.-h. (2019), The paradox of re-oligotrophication: the role of bottom-up versus top-down controls on the phytoplankton community. *Oikos*, 128: 1666-1677. <https://doi.org/10.1111/oik.06399>*

*Rémy D. Tadonléké, Jérôme Lazzarotto, Orlane Anneville, Jean-Claude Druart, Phytoplankton productivity increased in Lake Geneva despite phosphorus loading reduction, *Journal of Plankton Research*, Volume 31, Issue 10, October 2009, Pages 1179–1194, <https://doi.org/10.1093/plankt/fbp063>*

We thank the reviewer for asking to provide more information on this interesting point. The zooplankton was unlikely to control the *Uroglena* bloom. Such a lack of control from zooplankton can be explained by two reasons. Firstly, the low abundance of zooplankton and therefore its reduced effectiveness in general in controlling the phytoplankton in Lake Geneva (Tadonleke et al. 2009; Anneville et al. 2019). Secondly, and probably the main reason, is that *Uroglena* is not heavily consumed (Lehman and Sandgren 1985) due to its size, colonial form, and the presence of mucus. This aspect is now being discussed in the manuscript:

L236-L245:

“Phytoplankton control through zooplankton grazing (Sommer et al. 1986; 2012) was also examined. The zooplankton ^{14}C signatures were similar to the bloom signature in the open water (Table S2) hinting that zooplankton grazing on the bloom cannot be excluded. Our measurements suggest that *Daphnia* might have taken advantage of the massive phytoplankton biomass but was not able to control it. This is because of the limited strength of such a top-down control in Lake Geneva (Tadonleke et al. 2009) since herbivorous zooplankton have strongly declined following the increase in zooplanktivorous fish (Anneville et al. 2019). The limited control of *Uroglena* development by zooplankton could also be explained by the morphological characteristics of *Uroglena*, e.g., its size, colonial form and the presence of mucilage, which make it poorly vulnerable to zooplankton grazing (Lehman and Sandgren 1985).”

2. The qualitative and visual match between the particle tracking results in Figure 2A and the satellite observations of the bloom in Figure 1 are quite impressive. In addition, I appreciate the hydrodynamic explanation of the large geographical extent of the bloom. However, I'm having a hard time assessing whether or not these results are physically consistent. Part of the concern is that there is no assessment of the 3D hydrodynamic model validity and the details of the model configuration are lacking. For example, the model stratification and currents are not compared with observations. In addition, there is not a description of the advective time scales that might be expected during the simulation period. Is it possible to add an assessment of the model currents and stratification, perhaps at the LéXPLORE station, to the Supplementary Information? That addition would give some confidence in the model's ability to predict the advection and dispersion of the bloom, beyond a qualitative, graphical comparison, especially where there is some uncertainty with respect to the start date of the bloom.

We are grateful for the Reviewer's appreciation of our work. We agree that the validation and evaluation of the 3D hydrodynamic model should have been explained in more details in the previous version of the manuscript. We now document the performance of the model with four kinds of validation:

(i) Long-term calibration and validation of the model

The current model configurations are directly adopted from Safin et al., (2022), in which they calibrated different model parameters (e.g., Smagorinsky viscosity and Dalton number) for Lake Geneva based on current velocity and temperature measurements at 1 and 4 stations, respectively. The analysis during 15 January–15 December 2019 indicated a temperature root mean square error (RMSE) of 0.95-1.45 [°C] across different monitoring stations, and a velocity RMSE of 0.033 [m s^{-1}] compared to the measurements at the LéXPLORE station (Figure 1). The calibration process involved Bayesian inference in conjunction with a smoothed particle Markov Chain Monte Carlo method. This is to the best of our knowledge the most advance (and computationally demanding) calibration procedure we have seen in the literature so far. We revised the text under “Methods → 3D numerical model” accordingly (L460-465).

We also added “*Text S1-Hydrodynamic model validation, and evaluation*” to the Supplementary Information, which includes information on the short-term validation

and evaluation of the model, which are briefly described under items (ii) and (iii) below. In doing so, we added another monitoring station, called “*Buchillon*” (red star in Figure 1), to the analysis presented in the manuscript.

(ii) Short-term validation and evaluation of the water current

We compared the velocity measurements at the *LéXPLORE* station (Figure 1) with the model results. An acoustic Doppler current profiler (ADCP) measures the horizontal velocity components of the water every 10 min during our simulation period, out of which the measurements between 11-30 m layer (the surface layer contributing mainly to the particle tracking and EOF analysis) were used for comparison with the model. Although there are some discrepancies between the details of measurements and model results, the model was able to capture the velocity variation after a spin up time of ~3 weeks (Figure S1). In particular, the events with high current velocities on 17-Aug and 27-Aug, and the diurnal variation during 31-Aug to 7-Sep (the period used in our analyses and particle tracking) are in a good agreement with the numerical simulation results. Furthermore, the distribution of velocity magnitudes and directions of the model are strongly correlated with the observations (Figure S2).

(iii) Short-term validation and evaluation of water temperature and thermal structure

The evolution of temperatures and thermal structures during August 18 to September 8 were also compared between in situ measurements and numerical models. Observations include temperature profiles from a thermistor chain at the *LéXPLORE* station (red square in Figure 1), and temperatures at 1 m depth at *Buchillon* station (red star in Figure 1). In general, the model correlates accurately with the temperatures at 1m depth at the *Buchillon* station with an underestimation of ~-0.5 °C (Figure S5). The model also adequately follows the short-term thermal stratification of observations with some discrepancies mainly before the bloom onset (Figures S3-S4 and Table S1).

iv) Good agreement between model particle tracking results and satellite observation

Our particle tracking results (Figures 2A and S11), as the direct products of 3D hydrodynamic model, resembled the observed bloom patterns from satellite (Figures 1 and S6). We think this is also an indirect, yet strong, validation of the model during our period of interest. It is mentioned and discussed in the main text.

L122-130:

“Numerical modeling resembled the main observed satellite patterns. Forward trajectories of particles released hourly near the southern shore from September 3 (estimated bloom onset from both satellite and in-situ observations) onwards suggest that the western and central mushroom-like structures originated from these littoral regions (Figure 2A; see also backward tracking results in Figures S6A and S6B). In contrast, backward tracking of particles released on September 6 in the eastern bloom structure revealed that pelagic waters had mostly been trapped in this area (Figure S6C), with only weak connections to the southern shore and the lake’s eastern end.”

3. Related to the comment above, the 3D hydrodynamic model is initialized from rest with a horizontally uniform temperature field approximately 1-month prior to the time period of

interest. Is a one-month spin-up period adequate? Does this give enough time for the temperature field and currents to adjust from the unphysical initial conditions? For example Cimadoribus et al (2018) used a spin up period of over 3 months. More explanation is needed to justify the relatively short spin up period.

Cimadoribus, A. A., Lemmin, U., Bouffard, D., & Barry, D. A. (2018). Nonlinear dynamics of the nearshore boundary layer of a large lake (Lake Geneva). Journal of Geophysical Research: Oceans, 123, 1016–1031. <https://doi.org/10.1002/2017JC013531>

We thank the Reviewer for this comment, and agree with them that the model spin up time should have been justified and discussed in the text. Therefore, we revised the text and added supportive analysis to the manuscript: “Methods → 3D numerical model” and “*Text S1-Hydrodynamic model validation, and evaluation*”.

A recent study in Lake Geneva using the same code, i.e., MITgcm, with slightly different model configurations showed that the model cold start is not relevant as long as (i) the initial stratification is reasonable, and (ii) the model responds adequately to the strong persistent wind stress (Reiss, Lemmin, and Barry 2023). Here, *the model was initialized from rest with a horizontally homogeneous temperature profiles on July 26, 2021 obtained from measurements at the LÉXPLORE station (L467-469)*. In addition, the model responded notably well to the strong wind events on August 17 (Figure S1) and the one between August 23-30 (Figures S1-S3). *Therefore, the comparison of the model results with observations indicated a spin up time of ~3 weeks (Figure S1)*. However, the results obtained 35 d after the model initialization were mainly used in our study (Period of interest in Figure S1B). *(L474-476)*.

4. Are the Lagrangian particles subject to any vertical motions (e.g. vertical mixing, vertical velocities, buoyancy or sinking)? Or are they advected horizontally by the two-dimensional model currents at the appropriate depth level?

The method employed here tracks passively the 3D Lagrangian motion of inert particles with no mass and no volume using the velocity field obtained from the hydrodynamic model (L481-483). Particles are advected both horizontally and vertically. Along the vertical direction, they are subject to current-driven up/downwellings, buoyancy and sinking, as these processes alter the vertical component of the flow. Turbulent diffusion is directly accounted for in the solution of the momentum equations by the MITgcm model, which provides the input flow field. However, since we are interested in the deterministic advection of water masses resolved by the hydrodynamic model, and how the latter shapes the observed algal bloom pattern, diffusion (e.g., by random particle displacements) was not considered.

We modified the text in the “Methods → Lagrangian particle tracking” section *(L480-L489)*.

Specific comments

1. Figure 2B: Do the arrows have units? If so, what are they? If not, I'm confused about how the EOF decomposition of the velocity field described in equation 7 relates to this figure given the units of the magnitude time series are m^3/s .

We thank the reviewer for raising this valid point. We used “horizontal velocity flux”, i.e., the vertically integrated horizontal velocity, instead of “horizontal velocity” for the EOF analysis, as **it is usually a more relevant quantity than the vertically averaged velocity for investigating the basin-scale transport in large lakes** (Cimatoribus, Lemmin, and Barry 2019) (514-515).

We corrected and revised the “Methods → Empirical Orthogonal Function (EOF) analysis” section, accordingly (511-524). The arrows in Figures 2B and S12 indicate the spatial structure of the EOF modes, i.e., denoted by E in eq. (8). Therefore, they are physically dimensionless, but the unit vector, added to plot now as the quiver key, corresponds to the magnitude and phase presented in the inset.

2. Figure 4C: I'm confused by the label “global radiation”. Is this really a global radiation time series? Wouldn't it be more relevant to show the local solar radiation conditions?

We thank the reviewer, and understand the confusion. However, both COSMO-1 data (used in 3D modeling) and in situ measurements at *Changins* station (used for the long-term statistical analyses) returns global radiation. It is more relevant than solar radiation as it includes both direct and diffuse radiation, i.e., the scattered solar radiation by the atmospheric aerosols and reflection by clouds. Global radiation is the actual amount of energy that reaches the earth surface, and therefore is a better parameter in our analysis.

We revised the text by using the right term, i.e., global radiation (L184, L211, L290).

3. Page 13, first paragraph: What does OLA stand for?

OLA stands for “Observatoire des LAcS”. It is the name of the observatory hosted by INRAE and University of Savoie Mont-Blanc.

It is now specified in the text.

L348-351:

“The lake has been monitored systematically by different organizations. Bi-weekly/monthly physical and biogeochemical data have been measured by the long-term Observatory and experimentation on lakes, i.e., *Observatoire des LAcS* (OLA) and the *Commission Internationale pour la Protection des Eaux du Léman* (CIPEL) at SHL2 (Figure 1) since 1963”.

Reviewer #2

This studies focuses on the dynamics of Uroglena Sp. algal blooms in the low-nutrient Lake Geneva. It puts forward an interesting sequence of events that can trigger the occurrence of such blooms, which, given the low nutrient concentration, in the lake are quite rare. The sequence involves heavy rainfall and wind-induced coastal upwelling to increase the nutrient concentration in the lake's top layer and hence trigger the bloom followed by a prolonged period of warm and calm weather to sustain the bloom's development. The authors nicely combine in-situ measurements, satellite remote sensing and hydrodynamic modelling to make their point.

Although I am not an expert in freshwater algal blooms and thus unable to thoroughly evaluate the novelty of this study, I found it to be well-constructed, clear, and compelling. I would therefore recommend its publication.

We are grateful for the Reviewer's appreciation of our work and their recommendation for publication. We also appreciate the comments and suggestions that have been addressed below and helped improve the paper.

My main concerns/recommendations are the following:

*1. While the sequence of events that the authors have identified can explain the occurrence of algal blooms in end-July 1999 and early-Sept. 2021, it is not clear how often the same sequence of events was observed in Lake Geneva *without* triggering and sustaining a bloom. I think the authors should be better discuss whether such « false positive » events occurred. On page 10, they mention the absence of a bloom in mid-July 2021 without providing much details about it. Were there other similar events?*

We appreciate this comment by the Reviewer. We have discussed the occurrence of each individual prerequisite for the *Uroglena* bloom (Figures 4A-C and S15A-C). We also estimated the bloom occurrence probability (Figures 4D and S15D). There is no "false negative", but we obtained one "false positive" occasion at around mid-July 2021. As mentioned in the text, **the bloom absence in mid-July 2021 remains unexplained (L279-280)**. Although we do not have any measurements to support it, but we speculated that **it is likely due to the relatively short time lag between the heavy rainfall event and the occurrence of other essential meteorological conditions (< 10 days in this case, compared to 6-8 weeks for the two observed blooms), which was probably insufficient for incorporating terrestrial organic matter into the microbial loop favoring *Uroglena* growth (Kimura and Ishida 1986) (L280-284)**. We must emphasize that this was the only "false positive" obtained during 1998-2022.

2. I also regret the lack of generalization to other oligotrophic lakes (like e.g. Lake Baikal or Lake Tahoe). Is there any indication that the conclusions of this study could be applied elsewhere or have indeed been observed elsewhere? I think that by broadening the scope of their study beyond Lake Geneva, the authors would make it stronger.

We appreciate the reviewer's comment, which helped us to improve the discussion section.

Our work presents the first evidence that unexpected blooms can develop in lakes as a result of successive, non-exceptional meteorological patterns. Here, we do not claim that *Uroglena* bloom will develop in other lakes due to the same sequence of meteorological forcing. Instead, our aim is to highlight the importance of carefully evaluating the sequence of meteorological events when studying ecosystem dynamics. We modified the text in order to discuss each of those prerequisites, as well as some other key findings, from a global perspective:

- Effect of climate change on the ecosystem dynamics: Climate change has been identified as one of main drivers (Ho, Michalak, and Pahlevan 2019; Ho and Michalak 2020; Anneville et al. 2015), yet without compelling correlation between air temperature or precipitation and algal blooms (L47-49). The intensification of extreme weather events, such as floods, droughts, and heat waves, are also expected due to climate change (Stott 2016). Such extreme events have been reported to influence the ecosystem dynamics (Stockwell et al. 2020; Richardson et al. 2019) (L304-306).
- The role of wind-induced upwelling: The impact of upwelling on the ecosystem dynamics is still unresolved. In case of Lake Geneva, coastal upwellings are common (Bouffard et al. 2018; Reiss et al. 2020; Reiss, Lemmin, and Barry 2022) and hence cannot solely explain the exceptional bloom observed in September 2021. Wind-induced upwelling events have been also reported in other large meso-oligotrophic lakes, e.g., Lake Baikal (Shimaraev et al. 2012), Lake Tahoe (Roberts et al. 2021; Schladow et al. 2004), and Lake Constance (Rinke et al. 2009), with some of those events leading to amplified C_{chl-a} (Grachev et al. 2021; Leigh-Abbott et al. 1978; Bouffard et al. 2018) (L204-208).
- Effect of heavy rainfall events on excessive terrestrial organic matter and nutrients loading: Similar to the Lake Geneva, such heavy precipitation-driven pulses of terrestrial inputs have been identified as a trigger of algal bloom in other lakes (Green and Hufhines 2017; Michalak et al. 2013; Bakker and Hilt 2016; Callieri et al. 2014) (L225-227). These heavy rainfall events not only can load excessive nutrients into the water body (e.g., the massive cyanobacteria bloom in Lake Erie (Michalak et al. 2013)), but also increased dissolved organic carbon (DOC) inputs can be beneficial for abundance of some *Chrysophytes* (e.g., *Uroglena* blooms in Lake Geneva and Lake Beaver (Green and Hufhines 2017)) (L319-322).
- Mixotrophy as a selective functional trait in low-nutrient systems: Our observations, among others (Keck et al. 2020; Domaizon, Viboud, and Fontvieille 2003), suggest mixotrophy as a selective functional trait. Mixotrophy might provide a strong advantage in oligotrophic lakes, particularly under extreme precipitation events (Climate Scenarios for Switzerland 2018; Tan et al. 2023) (L316-317). Co-occurrence networks of bacteria and eukaryotes have already been reported in low-nutrient lakes and suggest strong interactions within the microbial community (Mikhailov et al. 2019). Accordingly, *Chrysophytes* are known to involve a variety of species exhibiting a wide range of nutritional behaviors. Among them the taxa *Uroglena* outcompetes other phytoplankton more effectively during nutrient-stressed conditions (Reynolds 2006). Its capability to use bacteria as a substitutable P-source (Urabe, Gurung, and Yoshida 1999) offers this taxon a strong

competitive advantage over other species but implies particular conditions since the presence of bacteria is mandatory for the blooming of this taxa in a laboratory culture (Kimura and Ishida 1986) (L253-260). Nevertheless, in oligotrophic systems experiencing local phosphorus inputs, the growth of mixotrophic species can be outcompeted by toxic cyanobacteria as it has been observed in Lake Baikal (Tikhonova et al. 2022) (L323-325).

3. I also regret that the authors didn't go one step further in their modelling work of the lake by considering a biogeochemical model and assessing whether such a model produced a bloom when the sequence of events put forward by the authors is satisfied. I understand that models remain simplifications of reality that don't capture all the processes driving the lake's dynamics. They remain however very useful to test assumptions. By running such a biogeochemical model, the authors would have obtained a more quantitative understanding of the importance of each step in the sequence they put forward and be able to assess e.g. what amount nutrient input is needed to trigger the bloom and hence how long the coastal upwelling should be sustained to achieve that.

We understand Reviewer's comment and agree that a coupled three-dimensional hydrodynamic and biogeochemical model would help understanding and predicting future algae bloom. We are however too far from this at the moment. A recent coupled 1D hydrodynamic and water quality model could explain the large differences in primary production and nutrient concentrations between two consecutive years (2012 and 2013) mostly as a result of deep mixing in winter 2012 (Krishna et al. 2021). This approach could not be used in our study as we look at a single event (algal bloom) with dynamics linked to 3D processes (upwelling, advective processes, river inputs, etc). 3D water quality models have been tested for Lake Geneva but their applications remain limited to seasonally averaged processes to evaluate if spatio-temporal heterogeneities modify the lake ecological status according to the European water framework directive (WFD) (Soulignac et al. 2019), or the abundance of typically growing algae on Lake Geneva (Soulignac et al. 2018). Such a model was not parameterized for mixotrophic *Uroglena* taxa and would not have been useful for this study.

Looking at the recent reviews of water quality models in lakes (Soares and Calijuri 2021; Vinçon-Leite and Casenave 2019), we do not think that deterministic water quality procedures can resolve such specific algae bloom and prefer to keep our simpler approach combining hydrodynamic and particle tracking. A follow up investigation - not implemented and therefore not possible at the moment - would be to modify the Lagrangian particle tracking with growth and mortality rates. We prefer to focus on this approach in the future for the investigation of specific complex events.

We slightly modified the text by adding such potential investigation as an outlook of the current study:

L327-330:

"As next steps to improve such understanding, we suggest incorporating growth and mortality rates into the particle tracking model, and setting up a coupled three-

dimensional hydrodynamic and biogeochemical model. Hybrid deterministic and data-driven models also seem a promising alternative (Deyle et al. 2022)".

A more minor comment concerns the description of the Lagrangian particle tracking (LPT) model where the authors indicate that they neglect diffusion (p.17). That seems quite unrealistic as, even if advection of the bloom by the currents is main driving mechanism, the lake remains turbulent and diffusion is hence present. I'm wondering whether the authors are not making this assumptions because they are running the LPT model backward in time and are hence uncomfortable with the idea of reversing the sign of diffusion, which is of course unphysical. I would advise them to have a look at the following paper that nicely explains how to account for diffusion in backward simulations: <http://journals.ametsoc.org/doi/10.1175/JTECH1874.1>

We appreciate the Reviewer's comment. We confirm that backward particle tracking can be only done physically and mathematically without diffusion. We also thank the Reviewer for the informative reading suggestion. The combined backward-forward approach proposed by Batchelder (2006) can indeed help partially overcome the issues related to the inclusion of diffusion in backward simulations. However, we note that we also neglected the diffusion (e.g., by random particle displacement) in our forward particle tracking model. This is because we are interested in the deterministic advection by the flow resolved in the hydrodynamic model, and not in smaller-scale turbulent mixing. Our main goal by using Lagrangian particle tracking is to understand the effect of lake physical processes on the advection of the bloom in the lake. This objective, together with using a high-resolution model with 50 m horizontal and O(0.1-10m) vertical grids, justify neglecting diffusion in particle tracking. The same particle tracking model with the same assumption has been adopted and validated for Lake Geneva with models with even coarser grids. However, we acknowledge the vital role of diffusion in larger systems with higher spatial and temporal scales, e.g., oceanic waters with O(1km) grid resolution and model output frequency of 12-hr (Batchelder 2006). Such lower spatial and temporal resolutions, as compared to our analyses here, necessitate considering the diffusion term in the LPT model to yield realistic results.

We revised the description of our Lagrangian particle tracking model.

L480-489:

"This method tracks passively the 3D Lagrangian motion of inert particles with no mass and no volume using the velocity field obtained from the hydrodynamic model. In our Lagrangian simulations, diffusion (e.g., by random particle displacement) was neglected and particles trajectories were followed based on 3D advection solely. Given the bloom's visually distinguishable spatial patterns (Figure 1), the advection of water masses was indeed assumed as the dominant process in its dispersion. In order to reproduce those large-scale observed patterns, we followed the deterministic advection by the flow resolved in the hydrodynamic model while not being in smaller-scale turbulent mixing."

Reviewer #3

This manuscript presents a combined earth observation approach, including satellite remote sensing, 3d-hydrodynamic modeling, in-lake water quality and optical observation, and statistical modeling of meteorological data to explain the occurrence of an algal bloom event in Lake Geneva, Switzerland/France. The authors present sound methods for evaluating different components of the lacustrine processes contributing to the bloom event. The manuscript is very well-written and well-thought-out. However, part of the discussion reads like it belongs in the Results section (e.g., the information on the three sequences of meteorological forcing).

We are grateful for the Reviewer's appreciation of our work. We have addressed their concerns, and modified the manuscript accordingly. The details are given below.

Regarding the sequential weather events, we agree that the analyses presented in Figure 4 seem to belong to the "Results" section. However, in the current manuscript storyline it is in fact the results of testing of the "discussed" bloom formation hypothesis: "To assess this hypothesis, long-term (1998-2022) meteorological data during June-September from a station around the lake (*Changins* in Figure 1) were analyzed (Figure 4A-C)" (L273-275). The information provided in this subsection, i.e., "Discussion → The importance of sequential weather events: Elucidating the rare emergence of blooms in a low-nutrient lake" are used to discuss why the *Uroglena* bloom occurrence remains at such a low frequency. Therefore, we respectfully prefer to keep it in the "Discussion" section.

My only concern regarding this manuscript is the generalizability of the core findings. Meteorological forcing is the only novel contribution in this manuscript that has not been robustly validated.

Please see our answer below (response to the *Reviewer comment* on Page 12) which is a repetition of our answer to Reviewer #2.

The probability of bloom occurrence numbers need to be investigated further. What does a probability value less than the arbitrary value of 0.75 mean?

We thank the reviewer for raising this point. The choice of > 0.75 threshold, as mentioned in the text, is somehow arbitrary. This relatively high value corresponds to a probability of at least 0.9 for each of three prerequisites (L559-560). The obtained daily estimated probability during 1998-2022 (Figure 4D) indicates that this threshold does not result in any "false negative", and therefore is adequate for bloom occurrence identification. However, the long-term analysis showed one "false positive" occurrence during mid-July 2021. The bloom absence during this period is likely due to the relatively short time lag between the heavy rainfall event and the occurrence of other essential meteorological conditions (< 10 days in this case, compared to 6-8 weeks for the two observed blooms), which was probably insufficient for incorporating terrestrial organic matter into the microbial loop favoring *Uroglena* growth (Kimura and Ishida 1986) (L279-284). On a general note,

and by mathematical definition, the lower the value of P_n (bloom), the less chance there is for the bloom occurrence (L560-561).

We modified the text under “Methods → Long-term statistical analysis of meteorological data” (L558-561), and “Discussion → The importance of sequential weather events: Elucidating the rare emergence of blooms in a low-nutrient lake” (L276).

Is this approach lake-specific, or can it work elsewhere? The authors claim these methodologies and findings can be extended to comparable aquatic systems. However, it has yet to be demonstrated in a lake other than Lake Geneva. Can the meteorological patterns in another lake with similar trophic levels detect the occurrence of algal bloom events? Without an investigation like that, the findings presented here would be a case study.

We appreciate the reviewer's comment, which helped us to improve the discussion section.

Our work presents the first evidence that unexpected blooms can develop in lakes as a result of successive, non-exceptional meteorological patterns. Here, we do not claim that *Uroglena* bloom will develop in other lakes due to the same sequence of meteorological forcing. Instead, our aim is to highlight the importance of carefully evaluating the sequence of meteorological events when studying ecosystem dynamics. We modified the text in order to discuss each of those prerequisites, as well as some other key findings, from a global perspective:

- Effect of climate change on the ecosystem dynamics: Climate change has been identified as one of main drivers (Ho, Michalak, and Pahlevan 2019; Ho and Michalak 2020; Anneville et al. 2015), yet without compelling correlation between air temperature or precipitation and algal blooms (L47-49). The intensification of extreme weather events, such as floods, droughts, and heat waves, are also expected due to climate change (Stott 2016). Such extreme events have been reported to influence the ecosystem dynamics (Stockwell et al. 2020; Richardson et al. 2019) (L304-306).
- The role of wind-induced upwelling: The impact of upwelling on the ecosystem dynamics is still unresolved. In case of Lake Geneva, coastal upwellings are common (Bouffard et al. 2018; Reiss et al. 2020; Reiss, Lemmin, and Barry 2022) and hence cannot solely explain the exceptional bloom observed in September 2021. Wind-induced upwelling events have been also reported in other large meso-oligotrophic lakes, e.g., Lake Baikal (Shimaraev et al. 2012), Lake Tahoe (Roberts et al. 2021; Schladow et al. 2004), and Lake Constance (Rinke et al. 2009), with some of those events leading to amplified C_{chl-a} (Grachev et al. 2021; Leigh-Abbott et al. 1978; Bouffard et al. 2018) (L204-208).
- Effect of heavy rainfall events on excessive terrestrial organic matter and nutrients loading: Similar to the Lake Geneva, such heavy precipitation-driven pulses of terrestrial inputs have been identified as a trigger of algal bloom in other lakes (Green and Hufhines 2017; Michalak et al. 2013; Bakker and Hilt 2016; Callieri et al. 2014) (L225-227). These heavy rainfall events not only can

load excessive nutrients into the water body (e.g., the massive cyanobacteria bloom in Lake Erie (Michalak et al. 2013)), but also increased dissolved organic carbon (DOC) inputs can be beneficial for abundance of some *Chrysophytes* (e.g., *Uroglena* blooms in Lake Geneva and Lake Beaver (Green and Hufhines 2017)) (L319-322).

- Mixotrophy as a selective functional trait in low-nutrient systems: Our observations, among others (Keck et al. 2020; Domaizon, Viboud, and Fontvieille 2003), suggest mixotrophy as a selective functional trait. Mixotrophy might provide a strong advantage in oligotrophic lakes, particularly under extreme precipitation events (Climate Scenarios for Switzerland 2018; Tan et al. 2023) (L316-317). Co-occurrence networks of bacteria and eukaryotes have already been reported in low-nutrient lakes and suggest strong interactions within the microbial community (Mikhailov et al. 2019). Accordingly, *Chrysophytes* are known to involve a variety of species exhibiting a wide range of nutritional behaviors. Among them the taxa *Uroglena* outcompetes other phytoplankton more effectively during nutrient-stressed conditions (Reynolds 2006). Its capability to use bacteria as a substitutable P-source (Urabe, Gurung, and Yoshida 1999) offers this taxon a strong competitive advantage over other species but implies particular conditions since the presence of bacteria is mandatory for the blooming of this taxa in a laboratory culture (Kimura and Ishida 1986) (L253-260). Nevertheless, in oligotrophic systems experiencing local phosphorus inputs, the growth of mixotrophic species can be outcompeted by toxic cyanobacteria as it has been observed in Lake Baikal (Tikhonova et al. 2022) (L323-325).

A minor note:

The remote sensing method adopted here is often termed as a three-band algorithm. The algorithm does not represent the NIR peak (the term used here). Instead, the algorithm semi-analytically measures an optical proxy of chl-a absorption at 665 nm.

We thank the Reviewer for this remark. It is now called “Three-band reflectance index”. We revised the *colormap label in Figure 1*, as well as the corresponding texts in the manuscript (L76-77, L107, and L411).

References

- Anneville, Orlane, Chun-Wei Chang, Gaël Dur, Sami Souissi, Frédéric Rimet, and Chih-hao Hsieh. 2019. "The Paradox of Re-oligotrophication: The Role of Bottom-up versus Top-down Controls on the Phytoplankton Community." *Oikos* 128 (11): 1666–77. <https://doi.org/10.1111/oik.06399>.
- Anneville, Orlane, Isabelle Domaizon, Onur Kerimoglu, Frédéric Rimet, and Stéphan Jacquet. 2015. "Blue-Green Algae in a 'Greenhouse Century'? New Insights from Field Data on Climate Change Impacts on Cyanobacteria Abundance." *Ecosystems* 18 (3): 441–58. <https://doi.org/10.1007/s10021-014-9837-6>.
- Bakker, Elisabeth S., and Sabine Hilt. 2016. "Impact of Water-Level Fluctuations on Cyanobacterial Blooms: Options for Management." *Aquatic Ecology* 50 (3): 485–98. <https://doi.org/10.1007/s10452-015-9556-x>.
- Batchelder, Harold P. 2006. "Forward-in-Time-/Backward-in-Time-Trajectory (FITT/BITT) Modeling of Particles and Organisms in the Coastal Ocean*." *Journal of Atmospheric and Oceanic Technology* 23 (5): 727–41. <https://doi.org/10.1175/JTECH1874.1>.
- Bouffard, Damien, Isabel Kiefer, Alfred Wüest, Stefan Wunderle, and Daniel Odermatt. 2018. "Are Surface Temperature and Chlorophyll in a Large Deep Lake Related? An Analysis Based on Satellite Observations in Synergy with Hydrodynamic Modelling and in-Situ Data." *Remote Sensing of Environment* 209: 510–23. <https://doi.org/10.1016/j.rse.2018.02.056>.
- Callieri, Cristiana, Roberto Bertoni, Mario Contesini, and Filippo Bertoni. 2014. "Lake Level Fluctuations Boost Toxic Cyanobacterial 'Oligotrophic Blooms.'" Edited by Hans G. Dam and Hans G. Dam. *PLoS ONE* 9 (10): e109526. <https://doi.org/10.1371/journal.pone.0109526>.
- Cimatoribus, A. A., U. Lemmin, and D. A. Barry. 2019. "Tracking Lagrangian Transport in Lake Geneva: A 3D Numerical Modeling Investigation." *Limnology and Oceanography* 64 (3): 1252–69. <https://doi.org/10.1002/lno.11111>.
- Climate Scenarios for Switzerland. 2018. "CH2018 - Climate Scenarios for Switzerland." NetCDF, CSV, ASCII, Rdata. National Centre for Climate Services. <https://doi.org/10.18751/CLIMATE/SCENARIOS/CH2018/1.0>.
- Deyle, Ethan R., Damien Bouffard, Victor Frossard, Robert Schwefel, John Melack, and George Sugihara. 2022. "A Hybrid Empirical and Parametric Approach for Managing Ecosystem Complexity: Water Quality in Lake Geneva under Nonstationary Futures." *Proceedings of the National Academy of Sciences* 119 (26): e2102466119. <https://doi.org/10.1073/pnas.2102466119>.
- Domaizon, Isabelle, Sylvie Viboud, and Dominique Fontvieille. 2003. "Taxon-Specific and Seasonal Variations in Flagellates Grazing on Heterotrophic Bacteria in the Oligotrophic Lake Annecy – Importance of Mixotrophy." *FEMS Microbiology Ecology* 46 (3): 317–29. [https://doi.org/10.1016/S0168-6496\(03\)00248-4](https://doi.org/10.1016/S0168-6496(03)00248-4).
- Grachev, Mikhail, Yuriy Bukin, Vadim Blinov, Oleg Khlystov, Alena Firsova, Maria Bashenkaeva, Oxana Kamshilo, et al. 2021. "Is a High Abundance of Spring Diatoms in the Photic Zone of Lake Baikal in July 2019 Due to an Upwelling Event?" *Diversity* 13 (10): 504. <https://doi.org/10.3390/d13100504>.
- Green, W. Reed, and Bradley Hufhines. 2017. "A Rare Uroglena Bloom in Beaver Lake, Arkansas, Spring 2015." *Lake and Reservoir Management* 33 (1): 8–13. <https://doi.org/10.1080/10402381.2016.1238427>.

- Ho, Jeff C., and Anna M. Michalak. 2020. "Exploring Temperature and Precipitation Impacts on Harmful Algal Blooms across Continental U.S. Lakes." *Limnology and Oceanography* 65 (5): 992–1009. <https://doi.org/10.1002/lno.11365>.
- Ho, Jeff C., Anna M. Michalak, and Nima Pahlevan. 2019. "Widespread Global Increase in Intense Lake Phytoplankton Blooms since the 1980s." *Nature* 574 (7780): 667–70. <https://doi.org/10.1038/s41586-019-1648-7>.
- Keck, François, Laurent Millet, Didier Debroas, David Etienne, Didier Galop, Damien Rius, and Isabelle Domaizon. 2020. "Assessing the Response of Micro-Eukaryotic Diversity to the Great Acceleration Using Lake Sedimentary DNA." *Nature Communications* 11 (1): 3831. <https://doi.org/10.1038/s41467-020-17682-8>.
- Kimura, Bon, and Yuzaburo Ishida. 1986. "Possible Phagotrophic Feeding of Bacteria in a Freshwater Red Tide Chrysophyceae *Uroglena Americana*." *NIPPON SUISAN GAKKAISHI* 52 (4): 697–701. <https://doi.org/10.2331/suisan.52.697>.
- Krishna, Shubham, Hugo N. Ulloa, Onur Kerimoglu, Camille Minaudo, Orlane Anneville, and Alfred Wüest. 2021. "Model-Based Data Analysis of the Effect of Winter Mixing on Primary Production in a Lake under Reoligotrophication." *Ecological Modelling* 440 (January): 109401. <https://doi.org/10.1016/j.ecolmodel.2020.109401>.
- Lehman, John T., and Craig D. Sandgren. 1985. "Species-specific Rates of Growth and Grazing Loss among Freshwater Algae1." *Limnology and Oceanography* 30 (1): 34–46. <https://doi.org/10.4319/lo.1985.30.1.0034>.
- Leigh-Abbott, Mark R., John A. Coil, Thomas M. Powell, and Peter J. Richerson. 1978. "Effects of a Coastal Front on the Distribution of Chlorophyll in Lake Tahoe, California-Nevada." *Journal of Geophysical Research: Oceans* 83 (C9): 4668–72. <https://doi.org/10.1029/JC083iC09p04668>.
- Michalak, Anna M., Eric J. Anderson, Dmitry Beletsky, Steven Boland, Nathan S. Bosch, Thomas B. Bridgeman, Justin D. Chaffin, et al. 2013. "Record-Setting Algal Bloom in Lake Erie Caused by Agricultural and Meteorological Trends Consistent with Expected Future Conditions." *Proceedings of the National Academy of Sciences* 110 (16): 6448–52. <https://doi.org/10.1073/pnas.1216006110>.
- Mikhailov, Ivan S., Yulia R. Zakharova, Yuri S. Bukin, Yuri P. Galachyants, Darya P. Petrova, Maria V. Sakirko, and Yelena V. Likhoshway. 2019. "Co-Occurrence Networks Among Bacteria and Microbial Eukaryotes of Lake Baikal During a Spring Phytoplankton Bloom." *Microbial Ecology* 77 (1): 96–109. <https://doi.org/10.1007/s00248-018-1212-2>.
- Reiss, R. S., U. Lemmin, and D. A. Barry. 2022. "Wind-induced Hypolimnetic Upwelling between the Multi-depth Basins of Lake Geneva during Winter: An Overlooked Deepwater Renewal Mechanism?" *Journal of Geophysical Research: Oceans* 127 (6). <https://doi.org/10.1029/2021JC018023>.
- Reiss, R. S., U. Lemmin, A. A. Cimadoribus, and D. A. Barry. 2020. "Wintertime Coastal Upwelling in Lake Geneva: An Efficient Transport Process for Deepwater Renewal in a Large, Deep Lake." *Journal of Geophysical Research: Oceans* 125 (8). <https://doi.org/10.1029/2020JC016095>.
- Reiss, R. S., Ulrich Lemmin, and David Andrew Barry. 2023. "What Role Does Stratification Play during Winter in Wind-Induced Exchange between the

- Multi-Depth Basins of a Large Lake (Lake Geneva)?” *Journal of Great Lakes Research* 49 (2): 406–21. <https://doi.org/10.1016/j.jglr.2023.02.005>.
- Reynolds, C. S. 2006. *The Ecology of Phytoplankton*. 1st ed. Cambridge University Press. <https://doi.org/10.1017/CBO9780511542145>.
- Richardson, Jessica, Heidrun Feuchtmayr, Claire Miller, Peter D. Hunter, Stephen C. Maberly, and Laurence Carvalho. 2019. “Response of Cyanobacteria and Phytoplankton Abundance to Warming, Extreme Rainfall Events and Nutrient Enrichment.” *Global Change Biology* 25 (10): 3365–80. <https://doi.org/10.1111/gcb.14701>.
- Rinke, Karsten, Andrea M. R. Huber, Sebastian Kempke, Magdalena Eder, Thomas Wolf, Wolfgang N. Probst, and Karl-Otto Rothhaupt. 2009. “Lake-wide Distributions of Temperature, Phytoplankton, Zooplankton, and Fish in the Pelagic Zone of a Large Lake.” *Limnology and Oceanography* 54 (4): 1306–22. <https://doi.org/10.4319/lo.2009.54.4.1306>.
- Roberts, Derek C., Galen C. Egan, Alexander L. Forrest, John L. Largier, Fabian A. Bombardelli, Bernard E. Laval, Stephen G. Monismith, and Geoffrey Schladow. 2021. “The Setup and Relaxation of Spring Upwelling in a Deep, Rotationally Influenced Lake.” *Limnology and Oceanography* 66 (4): 1168–89. <https://doi.org/10.1002/lno.11673>.
- Safin, Artur, Damien Bouffard, Firat Ozdemir, Cintia L. Ramón, James Runnalls, Fotis Georgatos, Camille Minaudo, and Jonas Šukys. 2022. “A Bayesian Data Assimilation Framework for Lake 3D Hydrodynamic Models with a Physics-Preserving Particle Filtering Method Using SPUX-MITgcm V1.” *Geoscientific Model Development* 15 (20): 7715–30. <https://doi.org/10.5194/gmd-15-7715-2022>.
- Schladow, S. Geoffrey, Sveinn Ó. Pálmarrsson, Todd E. Steissberg, Simon J. Hook, and Fred E. Prata. 2004. “An Extraordinary Upwelling Event in a Deep Thermally Stratified Lake.” *Geophysical Research Letters* 31 (15): 2004GL020392. <https://doi.org/10.1029/2004GL020392>.
- Shimaraev, M. N., E. S. Troitskaya, V. V. Blinov, V. G. Ivanov, and R. Yu. Gnatovskii. 2012. “Upwellings in Lake Baikal.” *Doklady Earth Sciences* 442 (2): 272–76. <https://doi.org/10.1134/S1028334X12020183>.
- Soares, Laura Melo Vieira, and Maria do Carmo Calijuri. 2021. “Deterministic Modelling of Freshwater Lakes and Reservoirs: Current Trends and Recent Progress.” *Environmental Modelling & Software* 144 (October): 105143. <https://doi.org/10.1016/j.envsoft.2021.105143>.
- Sommer, Ulrich, Rita Adrian, Lisette De Senerpont Domis, James J. Elser, Ursula Gaedke, Bas Ibelings, Erik Jeppesen, et al. 2012. “Beyond the Plankton Ecology Group (PEG) Model: Mechanisms Driving Plankton Succession.” *Annual Review of Ecology, Evolution, and Systematics* 43 (1): 429–48. <https://doi.org/10.1146/annurev-ecolsys-110411-160251>.
- Sommer, Ulrich, Z. Gliwicz, W. Lampert, and A. Duncan. 1986. “The PEG-Model of Seasonal Succession of Planktonic Events in Fresh Waters.” *Archiv. Fur Hydrobiologie* 106 (January).
- Soullignac, Frédéric, Orlane Anneville, Damien Bouffard, Vincent Chanudet, Etienne Dambrine, Yann Guénand, Tristan Harmel, et al. 2019. “Contribution of 3D Coupled Hydrodynamic-Ecological Modeling to Assess the Representativeness of a Sampling Protocol for Lake Water Quality Assessment.” *Knowledge &*

- Management of Aquatic Ecosystems*, no. 420: 42.
<https://doi.org/10.1051/kmae/2019034>.
- Soullignac, Frédéric, Pierre-Alain Danis, Damien Bouffard, Vincent Chanudet, Etienne Dambrine, Yann Guénand, Tristan Harmel, et al. 2018. "Using 3D Modeling and Remote Sensing Capabilities for a Better Understanding of Spatio-Temporal Heterogeneities of Phytoplankton Abundance in Large Lakes." *Journal of Great Lakes Research* 44 (4): 756–64.
<https://doi.org/10.1016/j.jglr.2018.05.008>.
- Stockwell, Jason D., Jonathan P. Doubek, Rita Adrian, Orlane Anneville, Cayelan C. Carey, Laurence Carvalho, Lisette N. De Senerpont Domis, et al. 2020. "Storm Impacts on Phytoplankton Community Dynamics in Lakes." *Global Change Biology* 26 (5): 2756–84. <https://doi.org/10.1111/gcb.15033>.
- Stott, Peter. 2016. "How Climate Change Affects Extreme Weather Events." *Science* 352 (6293): 1517–18. <https://doi.org/10.1126/science.aaf7271>.
- Tadonleke, R, Jerome Lazzarotto, Orlane Anneville, and Jean-Claude Druart. 2009. "Phytoplankton Productivity Increased in Lake Geneva despite Phosphorus Loading Reduction." *Journal of Plankton Research* 31 (10): 1179–94.
<https://doi.org/10.1093/plankt/fbp063>.
- Tan, Xuezhi, Xinxin Wu, Zeqin Huang, Jianyu Fu, Xuejin Tan, Simin Deng, Yaxin Liu, Thian Yew Gan, and Bingjun Liu. 2023. "Increasing Global Precipitation Whiplash Due to Anthropogenic Greenhouse Gas Emissions." *Nature Communications* 14 (1): 2796. <https://doi.org/10.1038/s41467-023-38510-9>.
- Tikhonova, Irina, Anton Kuzmin, Galina Fedorova, Ekaterina Sorokovikova, Andrey Krasnopeev, Anastasia Tsvetkova, Yulia Shtykova, et al. 2022. "Toxic Cyanobacteria Blooms of Mukhor Bay (Lake Baikal, Russia) during a Period of Intensive Anthropogenic Pressure." *Aquatic Ecosystem Health & Management* 25 (4): 85–97. <https://doi.org/10.14321/ae hm.025.04.85>.
- Urabe, J, Tb Gurung, and T Yoshida. 1999. "Effects of Phosphorus Supply on Phagotrophy by the Mixotrophic Alga *Uroglena Americana* (Chrysophyceae)." *Aquatic Microbial Ecology* 18: 77–83. <https://doi.org/10.3354/ame018077>.
- Vinçon-Leite, Brigitte, and Céline Casenave. 2019. "Modelling Eutrophication in Lake Ecosystems: A Review." *Science of The Total Environment* 651 (February): 2985–3001. <https://doi.org/10.1016/j.scitotenv.2018.09.320>.

19th Jan 24

Dear Dr Irani Rahaghi,

Your manuscript titled "Combined earth observations illuminate the path to a freshwater algal bloom emergence" has now been seen by our reviewers, whose comments appear below. In light of their advice we are delighted to say that we are happy, in principle, to publish a suitably revised version in Communications Earth & Environment under the open access CC BY license (Creative Commons Attribution v4.0 International License).

We therefore invite you to revise your paper one last time to address the remaining concerns of our reviewers. In particular, we require that you revise your manuscript to make the specificity of your analysis and findings to this single event in Lake Geneva clear throughout. At the same time we ask that you edit your manuscript to comply with our format requirements and to maximise the accessibility and therefore the impact of your work.

EDITORIAL REQUESTS:

*****Please take care to match our formatting and policy requirements. We will check revised manuscript and return manuscripts that do not comply. Such requests will lead to delays. *****

SUBMISSION INFORMATION:

OPEN ACCESS:

Communications Earth & Environment is a fully open access journal. Articles are made freely accessible on publication under a CC BY license (Creative Commons Attribution 4.0 International License). This license allows maximum dissemination and re-use of open access materials and is preferred by many research funding bodies.

For further information about article processing charges, open access funding, and advice and support from Nature Research, please visit <https://www.nature.com/commsenv/article-processing-charges>

At acceptance, you will be provided with instructions for completing this CC BY license on behalf of all authors. This grants us the necessary permissions to publish your paper. Additionally, you will be asked to declare that all required third party permissions have been obtained, and to provide billing information in order to pay the article-processing charge (APC).

[link redacted]

Best regards,

José Luis Iriarte Machuca, PhD
Editorial Board Member
Communications Earth & Environment

Clare Davis, PhD
Senior Editor
Communications Earth & Environment

www.nature.com/commsenv/
@CommsEarth

REVIEWERS' COMMENTS:

Reviewer #1 (Remarks to the Author):

The authors have addressed all comments and I'm happy to recommend publication in Communications Earth and Environment.

Reviewer #2 (Remarks to the Author):

I'm happy with the revisions made by the authors and don't have any additional comments.

Reviewer #3 (Remarks to the Author):

I would like to thank the authors for addressing my questions/concerns from the first review. I believe, the explanation provided to rebut my comment regarding the sequence of meteorological events can be improved. The authors rebutted the point as below.

“Here, we do not claim that Uroglena bloom will develop in other lakes due to the same sequence of

meteorological forcing. Instead, our aim is to highlight the importance of carefully evaluating the sequence of meteorological events when studying ecosystem dynamics.”

However, in the abstract, introduction, and discussion, they have linked the sequence of events to the bloom-triggering mechanism:

Lines 33-36: “We show that a specific sequence of meteorological conditions triggered this rare algal bloom: heavy rainfall promoting excessive organic matter and nutrients loading, followed by wind-induced coastal upwelling, and a prolonged period of warm, calm weather.”

Lines 67-70: “Our multidisciplinary approach provided novel insights into the causation of this exceptional bloom, highlighting the role of the “right” timing of the atmospheric forcing and the subsequent hydrodynamic processes.”

Lines 297-300: “Our study highlights the crucial role of meteorologically-driven processes and their timing in triggering algal blooms in a meso-oligotrophic lake, and that the future development of blooms do not simply follow trends in atmospheric forcing but are also closely linked to the dynamic sequence of meteorological events.”

I am not expecting that a set of conditions can predict a specific type of phytoplankton bloom elsewhere. Here, the basic expectation is that the meteorological conditions should trigger a bloom event or an increased biomass condition above a baseline concentration. If it cannot do that, I am unsure how to use this information. Excluding the false positive in July 2021, The probability of occurrence for one event in 2015 is very close to the threshold. Therefore, I encourage the authors to:

- Revise the explanation regarding the sequence of meteorological events: The authors can revisit their explanation in response to the comment regarding the sequence of meteorological events. They can provide more details and examples to support their argument that the sequence of events is not a generalizable bloom trigger condition. This will help the readers to understand their point of view and avoid any confusion.
- Revisit the statistical analysis: The authors can reconsider the lag windows for the meteorological data (E.g., eq. 11). That could help to develop a more generalizable bloom trigger condition.
- Else, please clarify the scope of the study: To avoid confusion, the authors can define the scope of their research in the abstract, introduction, and discussion. They can state that their study pertains to the triggering mechanism of a specific algal bloom event in a particular lake and provide a detection success rate of the trigger condition. This will help readers to understand the limited scope of the study and avoid any misinterpretations.

Response to reviewers

Manuscript ID: COMMSENV-23-1428-T

Title: *“Combined Earth observations illuminate the path to a freshwater algal bloom emergence”*

Authors: *Abolfazl Irani Rahaghi, Daniel Odermatt, Orlane Anneville, Oscar Sepúlveda Steiner, Rafael Sebastian Reiss, Marina Amadori, Marco Toffolon, Stéphan Jacquet, Tristan Harmel, Mortimer Werther, Frédéric Soullignac, Etienne Dambrine, Didier Jézéquel, Christine Hatté, Viet Tran-Khac, Serena Rasconi, Frédéric Rimet, Damien Bouffard*

This document follows the following fonts and colors:

- *Reviewer comment*
- Authors response
- **Quote from the revised manuscript**
- *L#: Line numbers in the revised manuscript without track-changes*

Reviewer #1

The authors have addressed all comments and I'm happy to recommend publication in Communications Earth and Environment..

We are grateful for the Reviewer's verification of our modified manuscript, and for their recommendation for publication.

Reviewer #2

I'm happy with the revisions made by the authors and don't have any additional comments.

We are grateful for the Reviewer's verification of our modified manuscript.

Reviewer #3

I would like to thank the authors for addressing my questions/concerns from the first review. I believe, the explanation provided to rebut my comment regarding the sequence of meteorological events can be improved.

We are grateful for the Reviewer's comment. We have addressed their concerns, and modified the manuscript accordingly. The details are given below.

The authors rebutted the point as below. "Here, we do not claim that Uroglena bloom will develop in other lakes due to the same sequence of meteorological forcing. Instead, our aim is to highlight the importance of carefully evaluating the sequence of meteorological events when studying ecosystem dynamics."

However, in the abstract, introduction, and discussion, they have linked the sequence of events to the bloom-triggering mechanism:

Lines 33-36: "We show that a specific sequence of meteorological conditions triggered this rare algal bloom: heavy rainfall promoting excessive organic matter and nutrients loading, followed by wind-induced coastal upwelling, and a prolonged period of warm, calm weather."

Lines 67-70: "Our multidisciplinary approach provided novel insights into the causation of this exceptional bloom, highlighting the role of the "right" timing of the atmospheric forcing and the subsequent hydrodynamic processes."

Lines 297-300: "Our study highlights the crucial role of meteorologically-driven processes and their timing in triggering algal blooms in a meso-oligotrophic lake, and that the future development of blooms do not simply follow trends in atmospheric forcing but are also closely linked to the dynamic sequence of meteorological events."

I am not expecting that a set of conditions can predict a specific type of phytoplankton bloom elsewhere. Here, the basic expectation is that the meteorological conditions should trigger a bloom event or an increased biomass condition above a baseline concentration. If it cannot do that, I am unsure how to use this information. Excluding the false positive in July 2021, The probability of occurrence for one event in 2015 is very close to the threshold.

We agree that the specificity of our analysis to the *Uroglena* blooms in Lake Geneva must be mentioned through the manuscript. We revised the text accordingly (L34-36; L65-73; L310-314).

We also agree that the probability of occurrence of one event in 2015 seems significant. However, we must emphasize that its probability of < 0.7 is much lower of the *Uroglena* bloom events, i.e., > 0.9 . Such a relatively low probability means that at least one of the three prerequisites (calm and warm conditions in this specific case) is not significant, which also explains the lack of an observed bloom in 2015. We also added such information to the manuscript (L292-296):

For example, the estimated bloom occurrence probability in early-June 2015, i.e., ~ 0.7 , is also close to the arbitrary threshold. However, no bloom has been reported in the lake during this period. This can be due to at least one insignificant prerequisite, e.g., lack of warm and calm conditions in this specific example from 2015.

Therefore, I encourage the authors to:

- *Revise the explanation regarding the sequence of meteorological events: The authors can revisit their explanation in response to the comment regarding the sequence of meteorological events. They can provide more details and examples to support their argument that the sequence of events is not a generalizable bloom trigger condition. This will help the readers to understand their point of view and avoid any confusion.*

As mentioned above, we now specify that the obtained results regarding the sequence of the meteorological conditions as the bloom trigger is for the current study, i.e., *Uroglena* blooms in Lake Geneva (L34-36; L65-73; L310-314).

Revisit the statistical analysis: The authors can reconsider the lag windows for the meteorological data (E.g., eq. 11). That could help to develop a more generalizable bloom trigger condition.

We appreciate the reviewer's comment. However, we must mention that the assumed time lags (60 days for heavy rainfall and 7 days for strong wind) and time lead (7 days for calm & warm conditions) are higher than those obtained for 2021 bloom in Lake Geneva, and therefore, the statistical model includes a safety factor for capturing favorable conditions. We modified the text to reflect this point.

L571-577:

Here, the time lags (for “post-precipitation” and “strong wind”) and time lead (for “calm & warm”) of 60 days, 7 days, and 7 days, were assumed for the three

prerequisites, respectively, indicated by j in equations 10-12. These time lags and time lead are slightly higher than those found for 2021 Uroglena bloom in Lake Geneva, i.e., ~ 6 weeks for heavy rainfall, ~ 4 days strong wind, and ~ 4 days for calm & warm conditions. This will give more flexibility to the statistical model, i.e., more days will be attributed with favorable prerequisites.

Else, please clarify the scope of the study: To avoid confusion, the authors can define the scope of their research in the abstract, introduction, and discussion. They can state that their study pertains to the triggering mechanism of a specific algal bloom event in a particular lake and provide a detection success rate of the trigger condition. This will help readers to understand the limited scope of the study and avoid any misinterpretations.

We thank the Reviewer for this suggestion. We revised the text accordingly (L34-36; L65-73; L310-314).